# Impacts of anthropogenic inputs on the hypoxia and oxygen dynamics in the Pearl River Estuary

Bin Wang [1,3], Jiatang Hu [1,2,*], Shiyu Li [1,2,*], Liuqian Yu [3], Jia Huang [1]

[1] School of Environmental Science and Engineering, Sun Yat-Sen University, Guangzhou, 510275, China

[2] Guangdong Provincial Key Laboratory of Environmental Pollution Control and Remediation Technology, Guangzhou 510275, China

[3] Department of Oceanography, Dalhousie University, Halifax, Nova Scotia, Canada

Correspondence to: Jiatang Hu (hujtang@mail.sysu.edu.cn), Shiyu Li (eeslsy@mail.sysu.edu.cn)

**Abstract.** In summer, the Pearl River Estuary (PRE) experiences hypoxia, largely driven by the high input of freshwater with low dissolved oxygen (DO) and abundant nutrients and particulate organic carbon from the Pearl River network. In this study, we used a well-validated physical-biogeochemical model together with a DO species tracing method to study the responses of hypoxia and oxygen dynamics to the anthropogenic perturbations of riverine inputs (i.e. DO, nutrients, and particulate organic carbon) in July-August 2006. Model results showed that hypoxia in the PRE was most sensitive to riverine inputs of particulate organic carbon, followed by DO concentrations and nutrients. Specifically, a 50% decrease (increase) in riverine input of particulate organic carbon led to a 47% decrease (64% increase) in hypoxic area, with the sediment oxygen demand and water column production being the two most important processes contributing to the changes in DO concentration. Changes in the riverine inputs of DO and nutrients had little impacts on the simulated hypoxia because of the buffering effects of re-aeration (DO fluxes across the air-sea interface), i.e. the re-aeration responded to the changes in surface apparent oxygen utilization (AOU) associated with river-induced variations of oxygen source and sink processes. The PRE features shallow waters (with averaged depth of 10 m) where oxygen provided by the re-aeration could penetrate to bottom waters via vertical diffusion and largely offset the changes in DO contributed by other oxygen source and sink processes. This study highlights the importance of re-aeration in reducing hypoxia variability in shallow estuaries.

## 1. Introduction

Recent decades have seen a decline in dissolved oxygen (DO) concentrations in most of the coastal oceans because of intensifying anthropogenic disturbances, leading to an increase in the occurrence and intensity of hypoxic conditions (Diaz and Rosenberg, 2008). Relations between the riverine nutrient loading and the hypoxic conditions (DO<2mg $L^{-1}$) have been well documented in many coastal hypoxic systems such as the Changjiang Estuary (Li et al., 2011; Ning et al., 2011), the Chesapeake Bay (Du and Shen, 2015; Hagy et al., 2004), and the northern Gulf of Mexico (NGOM) (Forrest et al., 2011; Justić et al., 2003). The classic paradigm for explaining the relations is that excessive nutrient inputs to the coastal oceans stimulate the high primary productivity there, and the subsequent decomposition of the organic matter in the bottom water consumes significant amount of DO that leads to hypoxia. As a result, nutrient reduction has been proposed to alleviate hypoxia in many hypoxic systems (e.g., the Chesapeake Bay (Scavia et al., 2006) and the NGOM (Justić et al., 2003)). Recent years have also seen an increasing number of studies showing that climate variation contributes to the spreading hypoxia in coastal oceans. The climate variation can change the ocean circulation or the vertical stratification to alter the balance between the oxygen source and sink processes (Rabalais et al., 2010). A modeling study conducted in the Chesapeake Bay has shown the good correlations between the climate variation, stratification, and the observed DO (Du and Shen, 2015). In addition, the global warming, as a symptom of climate variation, is another factor that can enhance the hypoxia. For example, Laurent et al. (2018) predicted a prolonged and more severe hypoxia in the northern Gulf of Mexico under a projected future (2100) climate state where the global warming leads to reduction in oxygen solubility and increased stratification.

(**Position of Figure 1**)

The Pearl River Estuary (PRE) is located on the Pearl River Delta (Figure 1a) and has a drainage area of 452,000 $km^2$. Previous studies have reported some summer hypoxic events in the PRE and explored the underlying mechanisms. Yin et al. (2004) suggests that stratification and estuarine circulations are two primary processes controlling the hypoxia in the PRE. Rabouille et al. (2008) compares the hypoxic conditions among four hypoxic systems and demonstrates the significance of tidal mixing to break hypoxia in the PRE. Zhang and Li (2010) further suggests that the contributions of biogeochemical processes to hypoxia in the PRE are also important. By conducting the oxygen balance analysis, they show that sediment oxygen demand (SOD) is the

dominant sink for oxygen. A more recent study by Wang et al. (2017) further points out that the balance of oxygen in the PRE is mainly controlled by the source and sink processes occurring in local and adjacent waters, among which the re-aeration (DO fluxes across the air-sea interface) and SOD determine the spatial distributions and durations of hypoxia in the PRE.

As a distinct river-dominated estuary, the PRE receives $3.3\times10^{11}$ m$^3$ yr$^{-1}$ of freshwater (Ou et al., 2009; Zhang and Li, 2010) along with a large amount of nutrients from the Pearl River network (Figure 1a), i. e. $5.6\times10^5$ t yr$^{-1}$ of dissolved inorganic nitrogen (DIN) and $9.9\times10^3$ t yr$^{-1}$ of dissolved inorganic phosphorus (DIP) (Hu and Li, 2009). Both dissolved inorganic nitrogen and phosphorus loadings have increased by about 60% from 1970 to 2000 and is predicted to increase by two times in 2050 due to the fast-growing agriculture and urbanization (Strokal et al., 2015). Understanding the response of hypoxia and oxygen dynamics to the changes in nutrient loading in the PRE is hence valuable for hypoxia prediction and management.

In addition to the nutrient loading, the particulate organic carbon (POC) are another important form of anthropogenic inputs that influence the hypoxia in the estuary ($\sim2.5\times10^6$ t yr$^{-1}$ from the Pearl River network (Zhang et al., 2013)). The POC can fuel the SOD when deposited and mineralized in the sediment layers, which has been found to dominate the DO depletions within the bottom waters of the PRE (Yin et al., 2004; Zhang and Li, 2010). In coastal systems the POC are often derived from the dead phytoplankton (Green et al., 2006), while in the PRE the POC mainly originate from the riverine inputs (Ye et al., 2017; Yu et al., 2010). This suggests the importance of studying the impact of riverine POC on hypoxia in the PRE.

In some cases, the hypoxia may also be induced by the advection of low-oxygen waters (Grantham et al., 2004; Montes et al., 2014; Wang, 2009; Wang et al., 2012). For example, Wang (2009) demonstrates that the hypoxia development in the Changjiang estuary is largely due to the Taiwan Warm Current bringing low–oxygen waters to the hypoxic zone. As to the PRE, the impact of riverine input of low-oxygen waters on hypoxia is also worth investigation considering the large amount of river discharge entering the estuary and that there has been hypoxia observed in its upper reaches (He et al., 2014).

Collectively the previous studies show that both natural and anthropogenic processes greatly contribute to hypoxia in the PRE. Understanding the respective roles of these two types of processes is important to faithfully predict future hypoxic events under the enhanced human activities and climate variations, which is useful for designing effective management strategies to prevent or remediate the hypoxic conditions in the PRE.

Here we focus on the role of human activities, i.e. different anthropogenic inputs, on hypoxia and oxygen dynamics in the PRE, whereas the role of natural processes will be reported in our future works. Specifically, we explore the impacts of varying anthropogenic inputs (riverine nutrients, POC, and DO) on hypoxia and oxygen dynamics in the PRE by using a three-dimensional (3D) coupled physical-biogeochemical model. The DO species tracing method introduced in Wang et al., (2017) is applied to isolate the effects of each oxygen source and sink process and to elucidate their interactions in this shallow and river-dominated estuarine system.

## 2.    Method

### 2.1   Model description and validation

#### 2.1.1 Model description

**Physical model**

Our physical model is a 1D-3D coupled model which incorporates the Pearl River network and the PRE (see Figure 1b, c for locations) into a single framework to resolve the dynamic interactions between these two regions (Hu and Li 2009). This coupled model was firstly developed with the biological and sediment models for the Pearl River-Estuary system to study the water, nutrients, and sediment flux budgets between the river network and estuary (Hu and Li 2009; Hu et al. 2011). Thereafter, it was extended to study the hypoxia (Wang et al. 2017) and the nutrient fluxes across the water-sediment interface (Liu et al., 2016) in the PRE.

The coupled model uses a so-called explicit coupling approach to incorporate the 1D model for the Pearl River Network and the 3D model for the PRE through the eight outlets (including Humen, Yamen, Hongqili, Hengmen, Modaomen, Jitimen, Hutiaomen, and Yamen; see Figure 1a for their locations). At each time step, the 3D model is forced by the simulated river discharges from the 1D model, and as a feedback, sends its simulated water levels at eight outlets to the 1D model as the downstream boundary conditions for the next time step. More detailed descriptions of the model methodology can be referred to Hu and Li (2009).

The cross-sectional integrated 1D model solves the Saint Venant equations of mass and momentum conservation by using a Preissmann implicit scheme and an iterative approach in the well-mixed river network. Figure 1b shows that the Pearl River Network is discretized into 1726 computational cross-sections, 189 nodes

(interactions between the different river branches), five upper boundaries (i.e. Shizui, Gaoyao, Shijiao, Laoyagang, and Boluo), and eight lower boundaries (the eight outlets). The upper boundaries of the 1D model are specified by the real-time observations of river discharges or water levels. The lower boundaries use the simulated water levels from the 3D model. Initial conditions are set to be zero for water levels and velocities and model time step is 5 seconds.

The 3D model is based on the Estuaries and Coastal Ocean Model with Sediment Module (ECOMSED; HydroQual Inc. (2002)) that has been extensively used to study the hydrodynamics in estuaries. The model has 183x186 horizontal grid cells with a resolution ranging from 400 m inside the Lingdingyang Bay to 4 km near the open boundaries (Figure 1c), and has 16 terrain-following sigma layers with refined resolution near the surface and bottom layers. The horizontal mixing is parameterized by a Smagorinsky-type formula (Smagorinsky, 1963) and the vertical mixing is calculated by the Mellor-Yamada level 2.5 turbulent closure model (Mellor and Yamada, 1982). The 3-D model is forced by the 6 hourly winds and 3 hourly surface heat fluxes from the ERA-interim (the Interim ECMWF Re-Analysis, http://www.ecmwf.int/en/research/climate-reanalysis/era-interim). Three open boundaries are specified by a monthly averaged profile of salinity and temperature (Hu and Li, 2009). Tides are introduced at the open boundaries using the water levels from the Oregon State University Tidal Data Inversion Software (OTIS). Freshwater inputs from the Pearl River network to the estuary use the river discharges simulated by the 1-D model.

The physical model is run from 1 November 2005 to 31 December 2006. More detailed descriptions and configurations can be found in Hu et al. (2011) and Wang et al. (2017).

**Biogeochemical model**

The biogeochemical model is the Row-Column AESOP model (RCA; HydroQual Inc. (2004)) that solves the mass balance equations for 26 state variables involved in five interactive cycles (i.e. the nitrogen cycle, the phosphorus cycle, the carbon cycle, the silicon cycle, and the oxygen dynamics). Interactions between these state variables with atmosphere and sediment are illustrated in Figure 2.

1                 (**Position of Figure 2**)

2     The equation of DO (mg $O_2$ $L^{-1}$) is given by:

$$\frac{\partial DO}{\partial t} + u\frac{\partial DO}{\partial x} + v\frac{\partial DO}{\partial y} + w\frac{\partial DO}{\partial z} - \frac{\partial}{\partial x}\left(E_x\frac{\partial DO}{\partial x}\right) - \frac{\partial}{\partial y}\left(E_y\frac{\partial DO}{\partial y}\right) - \frac{\partial}{\partial z}\left(E_z\frac{\partial DO}{\partial z}\right)$$

$$= WCP + REA - SOD \qquad\qquad (1)$$

where $x$ and $y$ represent the horizontal coordinates and $z$ the vertical coordinate; $u$, $v$, and $w$ (m $s^{-1}$) represent
velocity components in $x$, $y$, and $z$ coordinates, respectively; and $E_x$, $E_y$, and $E_z$ (m $s^{-2}$) are dispersion coefficients.
The velocity components and dispersion coefficients are computed by the physical model.
The term *WCP* represents the gross DO production rates in the water column (mg $O_2$ $L^{-1}$ $day^{-1}$), hereafter
the water column production, which is the combination of photosynthesis, respiration, nitrification, and
oxidation. Detailed equations for each component of the water column production can be seen in the Appendix
A. According to the DO budget analysis in Wang et al. (2017), the photosynthesis and respiration are two major
oxygen source and sink processes in the water column. Considering that photosynthesis and respiration are both
closely and directly correlated to the phytoplankton dynamics, they have the similar distributions and responses
to the external forcing. We therefore use the water column production to represent the net effects of water
column on the DO and hypoxia.
The term *REA* represents the re-aeration (mg $O_2$ $L^{-1}$ $day^{-1}$) at the air-sea interface, given as:

$$REA = k_a\theta_a^{T-20}(DO_{sat} - DO) \qquad\qquad (2)$$

where $DO_{sat}$ represents the DO concentration at saturation (mg $O_2$ $L^{-1}$) which is dependent on salinity and
temperature; $k_a$ is the surface mass transfer coefficient ($day^{-1}$); and $\theta_a$ is a temperature coefficient
(dimensionless). Values for these parameters can be seen in Table A2.
The term *SOD* represents the sediment oxygen demand (mg $O_2$ $L^{-1}$ $day^{-1}$) at the water-sediment
interface and $\Delta z$ represents thickness of the respective bottom grid cell (m).

$$SOD = \frac{s(DO - DO_{\text{sed}})}{\Delta z} \qquad (3)$$

where $s$ represents the transfer coefficient between the sediment and overlying water (m day$^{-1}$); $DO_{\text{sed}}$ represents DO concentrations (mg $O_2$ L$^{-1}$) in the sediment layers. In the RCA, a sediment flux module is incorporated to simulate the depositional flux of particulate organic matter (i.e. particulate organic carbon, particulate organic nitrogen, and particulate organic phosphate), the diagenesis processes in the sediment, and the transport of nutrients and DO from the sediment to the overlying water (Figure 2). Detailed descriptions about the sediment flux module can be seen in the Appendix B.

The simulation period for our biogeochemical model is the same as the physical model. Initial conditions were obtained from a two-month spin-up simulation which was repeated for three times to reach a steady state. River boundary conditions of biogeochemical variables were derived from the monthly observations in 2006 collected by the State Oceanic Administration (including nutrients and DO) and from a previous study (including different classes of dissolved organic carbon, particulate organic carbon, dissolved organic nitrogen, particulate organic nitrogen, dissolved organic phosphorus, and particulate organic phosphorus) (Liu et al., 2016). The open boundary conditions of biogeochemical variables were specified following Zhang and Li (2010).

### 2.1.2    Model validation

The physical-biogeochemical model has been validated against available observations during the July of 1999 in Hu and Li (2009) and July-August 2006 in Wang et al. (2017). We briefly summarize the validation results in 2006 below.

Being the coupling interface between the 1D model and the 3D models, the eight outlets serve as the lower boundaries of the 1D model and the river boundaries of the 3D model. It follows that the simulation of eight outlets is of great importance to the robustness of the 1D-3D coupled model. Model-data comparisons of water levels were conducted for eight stations including six outlets (i.e. Jiaomen, Hengmen, Modaomen, Jitimen, Hutiaomen, Yamen) and two other stations (i.e. Zhuhai and Wanshan) in Wang et al. (2017), with locations of the stations in their Figure 3. The normalized root-mean-square difference (RMSD) of water levels falls within 0.30 of the standard deviation of the observations and the correlation coefficient between the simulated and

observed water levels exceeds 0.95. This indicates that the coupled physical model is able to resolve the interactions between the river network and the estuary well. In addition, the tidal variations and the spring-neap tidal cycles in the PRE are well reproduced.

The PRE is characterized by the large extended river plume in the summer. Therefore, the model simulated salinity and temperature fields were validated against 146 profiles of salinity and temperature collected by estuary-wide monitoring cruise. The comparisons show small normalized RMSDs (both <0.60 of standard deviations of observations) and high correlations (>0.90 for salinity and >0.80 for temperature) between the model and observations, indicating that the coupled physical model is robust to reproduce the broad-scale features and intra-seasonal patterns of the main hydrodynamic features in the PRE.

For validation of biogeochemical fields, the simulated DO concentrations were validated against 53 oxygen profiles collected at 4 different cruises and distributed estuary-wide. The point to point comparisons show that the simulated DO concentrations agree well with observations, with the normalized RMSD below 0.8 standard deviation of the observations and the vast majority (85%) of the normalized errors falling within 1 standard deviation of the observations. Model-data comparisons of bottom DO concentrations further show that the model is able to reproduce the spatial distribution of the observed bottom DO and hypoxia. We have also assessed model skills in resolving source and sink processes associated with DO concentration. We found that the simulated spatial distributions and magnitudes of the re-aeration, respiration, and the SOD rates are similar with those of previous observational studies (see Table 3 in Wang et al. (2017)). The simulated chlorophyll-a, primary productivity and particulate organic carbon, which largely determine the respiration and the SOD rates (Zhang and Li, 2010), are also consistent with historical estimations. This suggests that our model is able to reproduce the oxygen dynamics properly.

In short, the model validation in Wang et al. (2017) indicate that our physical-biogeochemical model is robust to simulate the hydrodynamics and biogeochemical cycles in the PRE and is skillful in simulating summer hypoxia in 2006.

2.2 **The DO species tracing method**

The DO exhibits non-conservative behavior during the mixing in the estuary because of the oxygen source and sink processes described in Section 2.1.1 (Figure 3a). As shown in Figure 3b, the DO concentrations are

controlled by both the conservative (represented by the theory mixing curve) and the non-conservative effects (represented by the shading areas). The conservative effects are associated with physical advection and diffusion, while the non-conservative effects are due to the oxygen source and sink processes (i.e. re-aeration, the water column production, and the SOD). Quantifying the relative contributions of the respective effect is important to understand the DO dynamics during the mixing in the estuary. In a 0-D system, the non-conservative effects can be easily estimated as the products of time intervals and rates of corresponding source and sink processes. However, in a river and tide dominated estuary such as the PRE, this estimation is not straightforward because of the spatial connections of each source and sink process occurring in different locations. To address this problem, the DO species tracing method (referred to as the physical modulation method in Wang et al. (2017)) was introduced and implemented in our previous study to investigate the mechanisms of hypoxia in the PRE. By dividing the DO into different DO species, the tracing method can track the DO contributed by different source and sink processes. For example, Wang et al. (2017) found that about 28% of surface DO supplied by the re-aeration penetrated to the bottom waters and hence modulated the hypoxia in the PRE. In this study, the DO species tracing method is used to track contributions of each source and sink process to the DO dynamics and hypoxia under the different riverine inputs scenarios. Interactions between the oxygen source and sink processes will be investigated as well.

The DO species tracing method is incorporated into the biogeochemical model by explicitly including four numerical oxygen species as model tracers to track the DO contributed by the lateral boundary conditions ($DO_{BC}$ (mg $O_2$ $L^{-1}$)), air-sea re-aeration ($DO_{REA}$ (mg $O_2$ $L^{-1}$)), water column production ($DO_{WCP}$ (mg $O_2$ $L^{-1}$)), and SOD ($DO_{SOD}$ (mg $O_2$ $L^{-1}$)), respectively (Table A1). Equations of the four numerical oxygen species are given as below:

$$\frac{\partial DO_{BC}}{\partial t} + tran(DO_{BC}) = 0 \tag{4}$$

$$\frac{\partial DO_{REA}}{\partial t} + tran(DO_{REA}) = REA \tag{5}$$

$$\frac{\partial DO_{WCP}}{\partial t} + tran(DO_{WCP}) = WCP \tag{6}$$

$$\frac{\partial DO_{\text{SOD}}}{\partial t} + tran(DO_{\text{SOD}}) = SOD \qquad (7)$$

and

$$DO = DO_{\text{BC}} + DO_{\text{REA}} + DO_{\text{WCP}} - DO_{\text{SOD}} \qquad (8)$$

$$tran(DO) = tran(DO_{\text{BC}}) + tran(DO_{\text{REA}}) + tran(DO_{\text{WCP}}) - tran(DO_{\text{SOD}}) \qquad (9)$$

where $tran$ represents the physical transport processes, i.e. the advection $(u\frac{\partial}{\partial x} + v\frac{\partial}{\partial y} + w\frac{\partial}{\partial z})$ and diffusion $(-\frac{\partial}{\partial x}(E_{\text{x}}\frac{\partial}{\partial x}) - \frac{\partial}{\partial y}(E_{\text{y}}\frac{\partial}{\partial y}) - \frac{\partial}{\partial z}(E_{\text{z}}\frac{\partial}{\partial z}))$; $REA$ (mg $O_2$ L$^{-1}$ day$^{-1}$), $WCP$ (mg $O_2$ L$^{-1}$ day$^{-1}$), and $SOD$ (mg $O_2$ L$^{-1}$ day$^{-1}$) are the re-aeration, water column production, and SOD, which represent the net effects of the air-sea interface, the water column, and the water-sediment interface on the oxygen, respectively. Values of these terms are obtained from the biogeochemical model at each time step.

(**Position of Figure 3**)

According to the Eq. 4, the $DO_{\text{BC}}$ concentrations are only controlled by the advection and diffusion. By assigning the initial conditions and lateral boundary conditions of $DO_{\text{BC}}$ the same as those for DO, the mixing curve of $DO_{\text{BC}}$ will overlap the theory mixing curve shown in Figure 3b. It follows that the $DO_{\text{BC}}$ represents the conservative effects, while the $DO_{\text{REA}}$, $DO_{\text{WCP}}$, and $DO_{\text{SOD}}$ that include oxygen source or sink term represent the non-conservative effects.

The Eqs. 8 and 9 suggest that the DO concentration and its transport flux equal the sum of the concentrations and transport fluxes of the four DO species, respectively, the validity of which has been tested and confirmed in Wang et al. (2017). They show that there is little discrepancy between the DO concentrations calculated by Eq. (9) and Eq. (1), with 97% of the differences within the range of -2%~6% of the averaged DO concentrations. The hourly time series of domain-averaged DO calculated by the DO species tracing method also agree well with that calculated by the biogeochemical model with the R-square coefficient > 0.99 and the regression slope close to 1:1. In addition, the horizontal advective fluxes, vertical advective fluxes, and vertical

diffusive fluxes calculated by the tracing method are found to agree well with the respective fluxes calculated by the biogeochemical model, indicating that the tracing method is able to satisfactorily reproduce the physical transport processes of DO.

**2.3   Model experiments**

We conducted three groups of sensitivity experiments to study the response of hypoxia and oxygen dynamics to different scenarios of riverine inputs. Each group has two simulations, where the concentration of one type of the riverine inputs at eight river outlets is decreased and increased by 50%, respectively. These simulations are named as Base, RivDO-50%, RivDO+50%, RivNtr-50%, RivNtr+50%, RivPOC-50% and RivPOC+50%, with the basic information of each simulation presented in Table 1.

(**Position of Table 1**)

The Base simulation uses the realistic riverine inputs as mentioned in Section 2.1.1. In the Base simulation, the DO concentration in the Humen outlet, the largest river outlet in the PRE, is set to 4 mg L$^{-1}$ based on observations nearby. The RivDO-50% simulation where DO concentration from the eight outlets is decreased by 50% represents the scenario where hypoxia has developed in the Humen outlet, which has been reported in previous studies (e.g. He et al., 2014). In contrast, the RivDO+50% simulation, where the DO concentration from the eight outlets is increased by 50% to be close to that from the open boundaries, represents the scenario where the riverine input of DO is free from the anthropogenic impact. As to nutrient simulations, the RivNtr+50% and RivNtr-50% simulations increase and decrease nutrients concentrations from all eight outlets by 50%, respectively. The resulting riverine inputs in the two simulations will be close to the scenarios in 2050 and 1970 as reported by Strokal et al. (2015). Note that in the nutrient simulations, the concentrations of all nutrients (including dissolved silica, dissolved inorganic phosphorus, ammonia nitrogen, and nitrite and nitrate nitrogen) are set to vary at the same percentage that the effects of different combinations of changes in nutrients are not considered here. The hydrodynamic conditions are identical in all experiments.

The hypoxic extent in different simulations is quantified by the expected hypoxic area and hypoxic volume:

$$Hypoxic\ area = \sum p * \Delta s \tag{10}$$

$$Hypoxic\ volume = \sum p * \Delta v \tag{11}$$

where $\Delta s$, $\Delta v$, and $p$ are the area, the volume, and the hypoxic frequency of each grid cell. The hypoxic frequency $p$ is calculated by:

$$p = \frac{N_h}{N_s} * 100\% \tag{12}$$

where $N_h$ is the number of hours when hypoxia occurs, and $N_s$ is the total number of hours. In this study, the threshold of hypoxia is defined as 3 mg L$^{-1}$ (Luo et al., 2008; Rabalais et al., 2010).

## 3. Results

### 3.1 Hypoxia in the Pearl River Estuary

As shown in Figure 4, the hypoxia in the PRE starts to develop in April, peaks in August, and disappears in October, which is highly correlated ($R^2$=0.91) with the annual cycle of total river discharges with a time lag of one month. Figure 5 shows the model simulated DO distributions and hypoxic frequency in the bottom layer during the May-October. In May, the hypoxia is confined to the upstream of the Modaomen sub-estuary. In June, the bottom DO declines along the west coast of the PRE and the hypoxia starts to develop near the Gaolan island (see Figure 1 for its location). The hypoxia extends eastward to near the Hengqin Island in July and August. After August, the hypoxia retreats westward and almost disappears in October. Unlike the large spatial extent of hypoxia observed in the Changjiang Estuary (Wang, 2009; Wang et al., 2012) and the NGOM (Rabouille et al., 2008), the hypoxia in the PRE is confined to a small area as a result of the SOD and the re-aeration (Wang et al., 2017).

In 2006, oxygen observations are only available in July and August, which have demonstrated the occurrence of hypoxia. No observations are available for validating the model simulated hypoxia in other

months. We have collected and analyzed the oxygen observations from 1993 to 2009. However, the available observations are insufficient to resolve the annual cycle of the hypoxia in the PRE. To our knowledge, there are currently few studies on the annual cycle of hypoxia in the PRE due to the scarcity of observations. Discussions in this study therefore focus on the hypoxia in July-August when the distinct hypoxia was both observed and simulated by the model. Another motivation of focusing on July and August is that these two months are among the typical wet seasons in the PRE (Figure 4b), which is in line with our study on the effects of riverine inputs.

(**Position of Figure 4**)

(**Position of Figure 5**)

**3.2 Response of hypoxia and oxygen dynamics to riverine DO inputs**

Figure 6 shows the comparisons of bottom DO concentrations and hypoxic frequency during July-August for different DO simulations. In the RivDO-50% simulation, the spatial distribution of bottom DO is similar to that in the Base simulation except that hypoxia additionally occurs near the river outlets due to the inputs of low-oxygen waters from the upstream river network (Figure 6b, e). We have also examined the impact of reducing riverine DO in region farther away from the river outlets by excluding the hypoxic region near the river outlets. In this case the expected hypoxic area in RivDO-50% is only 2% higher than that in the Base simulation while the hypoxic volume is 26% higher (Figure 7a), indicating that the thickness of hypoxic water is greatly increased in RivDO-50%. In contrast, the RivDO+50% simulation yields higher bottom DO concentrations, leading to reductions of hypoxic area and volume by 23% and 30%, respectively (Figure 7a).

(**Position of Figure 6**)

(**Position of Figure 7**)

(**Position of Figure 8**)

Figure 7b, c, d further shows the changes in each DO species averaged over the bottom layer of the high frequency zone for different simulations in relative to the Base simulation. The high frequency zone here is defined as the area encompassed by the 10% isoline of July-August averaged hypoxic frequency and is denoted

by the white contour in Figure 8. To provide more insights into the response of different oxygen species to
riverine inputs, the spatial distributions of $DO_{BC}$ and $DO_{REA}$ in the bottom water are shown in Figure 8.
Differences in $DO_{WCP}$ and $DO_{SOD}$ concentrations between simulations are much smaller and hence omitted here.

4       Halving the riverine DO inputs in the RivDO-50% simulation yields lower $DO_{BC}$ concentrations but

higher $DO_{REA}$ in the bottom water (Figure 7b and Figure 8). The decrease in $DO_{BC}$ concentrations is largely
balanced by the increase in $DO_{REA}$ concentration in RivDO-50% simulation, which ultimately reduces the
magnitude of changes in hypoxic extent responding to the reduced riverine DO input. In the contrary, the
RivDO+50% simulation leads to higher $DO_{BC}$ concentrations (Figure 7b and Figure 8c) but lower $DO_{REA}$ in the
bottom water (Figure 7b and Figure 8f), which together reduces the net increase in bottom DO (Figure 7b).

10       The re-aeration buffering effects can be explained by the surface apparent oxygen utilization (AOU, the

difference between the actual DO concentration and its saturation at a known temperature and salinity). As
shown in Eq. (2), the re-aeration is a function of surface AOU. Halving the riverine DO inputs decreases the DO
concentrations in entire water column and therefore increases the surface AOU, which ultimately results in an
increase in re-aeration rate. In our model simulations, the surface domain-averaged saturated DO concentration
is ~7.42 mg $L^{-1}$, while the surface domain-averaged DO concentration in the Base and RivDO-50% simulations
are 6.81 and 6.57 mg $L^{-1}$, respectively. The surface AOU for the RivDO-50% simulation is 39% higher than that
for the Base simulation, which is consistent with the 38% increase in re-aeration rate for the RivDO-50%
simulation.
**3.3  Response of hypoxia and oxygen dynamics to riverine nutrient inputs**
As shown in Figure 7a, perturbing riverine nutrient inputs by 50% has relatively weak impact on hypoxic extent
(changes are within 10%). Among all the oxygen sink and source processes, the water column production and
re-aeration are the two that are most sensitive to variations in nutrient inputs. Halving the nutrient inputs by 50%
in the RivNtr-50% simulation remarkably reduces the primary productivity and water column production rates,
which in turn increases the surface AOU that facilitates the re-aeration. The increase in $DO_{REA}$ to the bottom
water via vertical diffusion offsets ~60% of the total DO loss associated with the reduced nutrient inputs in the
high hypoxic frequency zone (Figure 7c). As a result, the hypoxic area and hypoxia volume only increase by
about 10% in the RivNtr-50% simulation in relative to the Base simulation. In contrast, the RivNtr+50%

simulation yields higher water column production and lower re-aeration rate, with the changes of the two balance each other, and hence only leads to 4% and 6% decreases in hypoxic area and hypoxic volume, respectively in relative to the Base simulation.

### 3.4 Response of hypoxia and oxygen dynamics to riverine POC inputs

As shown in Figure 7, perturbing the riverine inputs of POC by 50% leads to significant changes in DO concentrations and hypoxic extent. In the RivPOC-50% simulation, the DO concentration increases by 0.56 mg L$^{-1}$ in the high hypoxic frequency zone and the hypoxic area and hypoxic volume decrease by 50% and 64%, respectively. In the contrary, RivPOC+50% simulation leads to significant decrease in the DO concentration, causing an extension of hypoxic area by 64% and a doubling of hypoxic volume (Figure 7a).

As to oxygen dynamics, the RivPOC-50% simulation leads to significant decline in the SOD rate (Figure 7d), and increase in the water column production rate (Figure 7d) as a result of the lower inputs of POC weakening the light attenuation in PRE. The combination of lower SOD and higher water column production rates increases oxygen concentration by 0.81 mg L$^{-1}$ in the bottom waters of the high hypoxic frequency zone (Figure 7d). However, decreasing the riverine inputs of POC in the RivPOC-50% simulation simultaneously weakens the re-aeration due to the decreased surface AOU. As a result, nearly 27% of the increased DO concentrations is offset by the decreased re-aeration in the high hypoxic frequency zone. In contrast, increasing the riverine inputs of POC in the RivPOC+50% simulation increases the SOD rates but weakens the water column production rates, which consequently reduces bottom water oxygen; nevertheless, 26% of the oxygen loss is offset by the enhanced re-aeration in this simulation.

To understand the impacts of changing the riverine inputs of POC on the water column production rates, we further examine how phytoplankton growth responds to varying riverine inputs of POC. The equation for the phytoplankton growth rate $G_P$ (day$^{-1}$) can be written as:

$$G_P = G_{Pmax} \cdot G(T) \cdot G(I) \cdot G(N) \tag{14}$$

where $G_{Pmax}$ represents the maximum growth rate at the optimum conditions (day$^{-1}$); $G(T)$, $G(I)$, and $G(N)$ represent the limitations by temperature, light, and the nutrients, respectively. These limitation coefficients are

non-dimensional scale values ranging from 0 to 1, with 0 representing no growth and 1 no limitation. The two POC simulations and the Base simulation have identical physical processes and hence same temperature limitation. Table 2 shows that changing the riverine inputs of POC has little impact on nutrient limitation but leads to large variations in light limitation, suggesting that the riverine inputs of POC can significantly affect the phytoplankton growth through light shading effects.

(**Position of Table 2**)

Considering the important role of re-aeration in POC simulations, we further quantify how re-aeration responds to the SOD by conducting a diagnostic analysis of $DO_{SOD}$ in July and August (Figure 9). Three vertical layers are defined: the upper layer (top 20% of the water column), middle layer (middle 60% of the water column), and bottom layer (20% of the water column above the sediment). Note that horizontal diffusion is omitted in the diagnostic analysis because its magnitude is much smaller than other terms. Diagnostic analysis of other DO species can be seen in Figures 11 and 12 of Wang et al. (2017). As shown in Figure 8, the SOD can affect the DO concentrations in the upper layer indirectly through the interactions with the vertical advection, the vertical diffusion, and the horizontal advection as explained below. First, the SOD consumes bottom DO by 0.53 mg L$^{-1}$ day$^{-1}$ and decrease the upward advective DO fluxes reaching the upper layer by 0.34 mg L$^{-1}$ day$^{-1}$. Second, the deoxygenation induced by SOD can increase the vertical DO gradient and facilitate the downward vertical diffusion of oxygen by 0.02 mg L$^{-1}$ day$^{-1}$ from the upper layer. Finally, the decreased upper DO concentrations affect the horizontal outfluxes of DO and ultimately result in a higher net horizontal advective flux by 0.21 mg L$^{-1}$ day$^{-1}$. Consequently, the net effect of the SOD on the upper DO is 0.15 mg L$^{-1}$ day$^{-1}$. which causes a decline of 2.22 mg L$^{-1}$ in DO concentrations in the surface layer. Figure 9 shows contributions of the SOD and the water column production rates to the changes of surface DO. The positive values of $\Delta(-DO_{SOD})$ and $\Delta(DO_{WCP})$ represent the increased DO concentrations due to the decrease of the SOD and increase of the water column production, respectively. In the RivPOC-50% simulation, decreasing the POC inputs decreases the SOD rate but increases the water column production rate, which in combine increase the DO concentrations in the surface layer. As a result, the re-aeration in the RivPOC-50% simulation is weakened, especially in the west of the lower estuary (Figure 10a).

1                  (Position of Figure 9)

2                  (Position of Figure 10)

**4. Discussion**
**4.1 Comparability of hypoxia in 2006**
In this study, we performed a series of numerical experiments together with the application of DO species
tracing method to study the effects of different anthropogenic inputs on hypoxia and oxygen dynamics in the
PRE. This study is the first attempt to quantitatively estimate the interactions between each DO source and sink
processes (e.g. DO buffering effects) under the anthropogenic perturbations in the PRE. The year 2006 was
selected because the distinct hypoxia was observed, and the available observations are relatively more abundant
than in other years. In addition, it is a wet year with the annual averaged total river discharge over 10,000 $\text{m}^3$ $\text{s}^{-1}$
(interannual variations of total discharges during 1999-2010 in the PRE can be seen in Figure s1 in the
supplement). Discussions are only focus on the hypoxia in July and August of 2006 when oxygen observations
are available. However, conclusions drawn here should be applicable to other years because previous studies
have reported similar locations and spatial extents of hypoxia in other years (Lin et al., 2001; Zhang and Li,
2010). The mechanisms underlying hypoxia of summer 2006 found here are also consistent with previous
studies on hypoxia in this region, such as the strong re-aeration (Zhang and Li, 2010), the dominance of the
SOD (Yin et al., 2004; Zhang and Li, 2010), and the important contributions of the allochthonous POC (Hu et
al., 2006; Yu et al., 2010).
**4.2 Relative contributions of different anthropogenic inputs**
Numerical experiments show that the hypoxia in the PRE is more sensitive to the riverine inputs of POC rather
than the nutrient loading (Figure 6a). This is distinct from other hypoxic systems such as the Chesapeake Bay
(Hagy et al., 2004) and the NGOM (Justić et al., 2003) that have observed close relation between nutrient
loading and hypoxia. We attribute this to the different characteristics of hypoxia in these systems (Table 3). In
the Chesapeake Bay, the dominant oxygen sink leading to hypoxia is the water column respiration, which is
associated with high primary productivity stimulated by the excessive nutrient loading (Hong and Shen, 2013).
In contrast, the bottom water DO depletions are dominated by the SOD in the NGOM (Murrell and Lehrter,
2011; Yu et al., 2015b) and the PRE (Yin et al., 2004; Zhang and Li, 2010). Hypoxia in the NGOM can be well

simulated with appropriate parameterization of SOD while neglecting the water column processes (Yu et al., 2015a).

However, the relative contributions of autochthonous POC (i.e. the POC generated by settling of phytoplankton after death) versus allochthonous POC to the SOD are different in the NGOM and the PRE. In the NGOM, the autochthonous POC serves as the major source of POC (Green et al., 2006), which means increasing the nutrient loading can facilitate the SOD by increasing the depositional fluxes of dead phytoplankton and ultimately promote the formation of hypoxia. In the PRE, the relative contributions of autochthonous versus allochthonous POC inputs to the SOD and hypoxia have long been a topic of debate. Some studies suggest that allochthonous POC dominate in wet seasons due to the high river discharges (Ye et al., 2017; Yu et al., 2010), while others argue that autochthonous inputs can also play an important role (Guo et al., 2015; Su et al., 2017). Previous studies (Guo et al., 2015; Hu et al., 2006; Ye et al., 2017) show that the ratios of allochthonous POC to autochthonous POC have distinct spatial and seasonal variabilities in the PRE. Generally, the allochthonous contributions dominate inside the estuary and gradually decrease seaward as the impact of the river discharges weakens (Hu et al., 2006; Jia and Peng, 2003). In our study, the high hypoxic frequency zone is near the Modaomen sub-estuary which receives high depositional fluxes of allochthonous POC. Therefore, the allochthonous inputs have dominant contributions to the SOD and summer hypoxia in the high hypoxic frequency zone.

(**Position of Table 3**)

The different POC sources in the NGOM and the PRE might be explained by their distinct physical and biogeochemical processes (Table 4). Firstly, the relative magnitudes of autochthonous versus allochthonous POC are different in the two hypoxic systems. The allochthonous inputs of POC in the NGOM and PRE are at the same magnitude: $3.8 \times 10^6$ t yr$^{-1}$ (Wang et al., 2004) and $2.5 \times 10^6$ t yr$^{-1}$ (Zhang et al., 2013), respectively. However, the autochthonous inputs in the two systems are different. According to our model results, the primary productivity in the PRE is $310.8 \pm 427.5$ mg C m$^{-2}$ day$^{-1}$, which is within the range of 183.9~1213 mg C m$^{-2}$ day$^{-1}$ reported by Ye et al., (2014). However, the observed primary productivity in the NGOM ranges from 330 to 7010 mg C m$^{-2}$ day$^{-1}$ (Quigg et al., 2011), the upper range of which is much higher than that in the PRE. The relatively lower primary productivity in the PRE is a result of the stronger phosphorus limitation (DIN:DIP ratio of 126 in the PRE versus 33 in the NGOM, respectively) and the light shading effects of high suspended sediment concentrations. The dominant role of the allochthonous POC in highly turbid estuaries have been

reported in previous studies (Fontugne and Jouanneau, 1987; Middelburg and Herman, 2007). Secondly, fates of
the allochthonous POC in the two systems are different due to the difference in the residence time between the
systems. In the PRE, the residence time is 3~5 days during the wet season, which is much shorter than in the
NGOM (~95 days). It follows that the allochthonous POC cannot be degraded completely and hence can
significantly fuel the SOD in the PRE. The difference in surface salinity distribution can also be used to explain
the different relative roles of allochthonous POC in the two hypoxic systems. Previous studies have suggested a
good correlation between the relative contributions of allochthonous POC and the salinity, namely the
contributions of allochthonous POC generally decrease as salinity increases seaward (Fontugne and Jouanneau,
1987; Middelburg and Herman, 2007). Similar correlations have also been reported in the PRE (Yu et al., 2010)
and NGOM (Wang et al., 2004). The surface salinity in the high hypoxia frequency zone varies between 0 to 10
psu during the wet season based on our model results, while the surface salinity in the hypoxic zone of the
NGOM is saltier than 24 psu even in the wet season according to the results from a well-validated physical
model in Yu et al. (2015a). This implies a more important role of allochthonous POC in the PRE than in the
NGOM. Finally, compositions of the allochthonous POC are different in the two hypoxic systems. Zhang and Li
(2010) mentioned that contributions of labile POC to the allochthonous POC are higher in the PRE than in the
NGOM.
(**Position of Table 4**)
**4.3 The importance of re-aeration in PRE**
Model results also highlight the importance of re-aeration in regulating DO dynamics and hypoxia migration in
the PRE. On the one hand, based on our previous study applying the same physical-biogeochemical model and
tracing method as here (Wang et al., 2017), the re-aeration together with the SOD are the most important
process controlling DO dynamics. Nearly 28% of the surface $DO_{REA}$ can reach the bottom layer, exerting a
strong constrain on the spatial extent and duration of hypoxia in the PRE. When turning off the re-aeration, the
high SOD will lead to a persistent hypoxia covering an area of over 3,000 $km^2$ in the PRE. On the other hand,
the re-aeration responds rapidly to the perturbations of riverine inputs, which moderates the DO changes
impacted by the perturbations. A conceptual diagram of these processes is illustrated in Figure 10. Compared
with other hypoxic systems, the re-aeration in the PRE is of great importance because of the shallow topography
and the strong re-aeration, which enable the surface oxygen supplied by re-aeration to penetrate to the bottom
water. Re-aeration thus can greatly influence spatial migration of hypoxia under the perturbations of riverine

inputs in the PRE. Furthermore, the shallow topography in the PRE allows the bottom SOD to indirectly affect the surface DO by decreasing the upward DO advective fluxes, which also facilitates strong re-aeration in the PRE. As we have described in section 3.3, the bottom SOD can lead to a decrease in surface DO concentrations by 2.22 mg $L^{-1}$. If turning off the SOD, the surface AOU would change from 0.61 to -1.61 mg $L^{-1}$, causing a change of re-aeration from 0.55 mg $L^{-1}$ $day^{-1}$ to -1.45 mg $L^{-1}$ $day^{-1}$. This indicates that the SOD could shift the role of re-aeration from a strong oxygen sink to a strong source.

(Position of Figure 11)

One counter-example to the shallow PRE is the NGOM, where the hypoxic zone is deeper such that the surface water and bottom hypoxic water is detached. Also, the observed SOD varies from 0.06 to 0.70 g $m^{-2}$ $day^{-1}$ in the summer season in the NGOM (Murrell and Lehrter 2011), which is much lower than those in the PRE (0.72~3.89 g $m^{-2}$ $day^{-1}$; Chung et al. (2004)). These characteristics together with the supersaturated DO concentrations in the surface water due to the high primary productivity make the re-aeration primarily an outgassing process in the NGOM (Yu et al., 2015b).

In the other hypoxic system, the Chesapeake Bay as described earlier, extended discussion on the importance of re-aeration is limited by a lack of observations and relevant studies of re-aeration. Nevertheless, according to our results, we can speculate that the re-aeration might be quite important in the Chesapeake Bay because the strong water column respiration can draw down the surface DO concentrations and enhance the re-aeration. However, the penetration of the oxygen supplied by re-aeration to the bottom layer is hard to be estimated without applying the DO species tracing method like our study or method similar in the Chesapeake Bay. In general, more relevant studies are required to examine the role of the re-aeration on hypoxia in the Chesapeake Bay.

## 5. Conclusion

This study uses a physical-biogeochemical model to simulate the DO dynamics and hypoxia in the PRE and investigate their responses to anthropogenic perturbations in riverine inputs. Model results based on simulation in 2006 shows that the hypoxia in the PRE starts in April, peaks in August, and disappears in October. Perturbing riverine inputs has strong impacts on DO dynamics and hypoxia. The hypoxic extent in the PRE is most sensitive to riverine input of particulate organic carbon, followed by oxygen and nutrients. This is different from other hypoxic systems (i.e. NGOM and Chesapeake Bay) because of the distinct physical and

biogeochemical features in the PRE, i.e. the shallow topography, high water exchange rates and dominance of
the SOD for DO depletions within bottom waters.
Model results also highlight the importance of re-aeration on hypoxia, which has strong buffering effects
on the oxygen dynamics in the PRE. River-induced changes in source and sink processes can trigger an opposite
shift in re-aerations by altering the surface AOU. In turn, the re-aeration can moderate the DO changes and
hypoxia shifts responding to the changes in the oxygen source and sink processes. The important role of
re-aeration in the PRE is due to the shallow waters and strong SOD in the estuary. Firstly, because of the
shallow topography, the SOD can affect the surface DO indirectly by decreasing the surface AOU and
consequently shifting re-aeration from an oxygen sink to a strong source process. Secondly, the shallow waters
enable the oxygen supplied by the re-aeration in to diffuse to bottom waters and compensate the DO loss by the
SOD.
**Appendix A: Each component of water column production**
The water column production (WCP) used in this study represents the net effects of water column on DO, which
is a combination of the photosynthesis, respiration, nitrification, and oxidation:
$$WCP = Phot - Resp - Nitrif - Oxid \tag{A1}$$
The first term *Phot* represents the photosynthesis (mg $O_2$ $L^{-1}$ $day^{-1}$):
$$Phot = \left[\alpha_{OC} \cdot \alpha_{NH_4} \cdot G_P \cdot P_c + \left(\alpha_{NO_{23}c}\right) \cdot \left(1 - \alpha_{NH_4}\right) \cdot G_P \cdot P_c\right] \tag{A2}$$
where $\alpha_{OC}$ represents oxygen to carbon ratio (mg $O_2$:mg C), $\alpha_{NH_4}$ represents the phytoplankton's preference
for ammonium uptake (dimensionless), $G_P$ represents specific phytoplankton growth rate ($day^{-1}$) which is
dependent on the temperature, light, and nutrients (including NO2+NO3, NH4, PO4, Si, see Eq. (14)), $P_c$
represents phytoplankton biomass (mg C $L^{-1}$), and $\alpha_{NO_{23}c}$ represents oxygen to carbon ratio for nitrate uptake
(mg $O_2$:mg N).
The term *Resp* represents the respiration (mg $O_2$ $L^{-1}$ $day^{-1}$):
$$Resp = \alpha_{OC} \cdot k_{PR}(T) \cdot P_c \tag{A3}$$
where the $k_{PR}(T)$ represents the temperature-dependent respiration rate ($day^{-1}$).
The term *Nitrif* represents the nitrification (mg $O_2$ $L^{-1}$ $day^{-1}$):

$$Nitrif = 2 \cdot \alpha_{\text{ON}} \cdot k_{14,15} \theta_{14,15}^{T-20} \cdot NH_4 \cdot \frac{DO}{K_{\text{nitri}} + DO} \tag{A4}$$

where $\alpha_{\text{ON}}$ represents the oxygen-to-nitrogen ratio (mg $O_2$:mg N), $k_{14,15}$ represents the nitrification rate at
20 °C ($day^{-1}$), $\theta_{14,15}$ represents the temperature coefficient (dimensionless), and $K_{\text{nitri}}$ represent the half
saturation constant for oxygen limitation (mg $O_2$ $L^{-1}$).
The term *Oxid* represents the oxidation of dissolved organic carbon, and dissolved sulfide (mg $O_2$ $L^{-1}$
$day^{-1}$):
$Oxid = \alpha_{\text{OC}} \cdot \left[ k_{20,0} \theta_{20,0}^{T-20} \cdot RDOC + k_{21,0} \theta_{21,0}^{T-20} \cdot LDOC \cdot \frac{LDOC}{K_{\text{LDOC}} + LDOC} + k_{22,0} \theta_{22,0}^{T-20} \cdot ReDOC \right.$
$\left. \cdot \frac{ReDOC}{K_{\text{LDOC}} + ReDOC} + k_{23,0} \theta_{23,0}^{T-20} \cdot ExDOC \cdot \frac{ExDOC}{K_{\text{LDOC}} + ExDOC} \right] \cdot \frac{P_{\text{c}}}{K_{\text{Pc}} + P_{\text{c}}} \cdot \frac{DO}{K_{\text{DO}} + DO}$
$+ k_{\text{O}_2^*} \theta_{\text{O}_2^*}^{T-20} \cdot O_2^* \cdot \frac{P_{\text{c}}}{K_{\text{Pc}} + P_{\text{c}}} \cdot \frac{DO}{K_{\text{DO}_{\text{O}_2^*}} + DO} \tag{A5}$
where $k_{20,0}$, $k_{21,0}$, $k_{22,0}$, $k_{23,0}$, and $k_{\text{O}_2^*}$ represent the oxidation rates of refractory dissolved organic carbon
(RDOC), labile dissolved organic carbon (LDOC), reactive dissolved organic carbon (ReDOC), algal exudate
dissolved organic carbon (ExDOC), and dissolved sulfide at 20 °C ($day^{-1}$); $\theta_{20,0}$, $\theta_{21,0}$, $\theta_{22,0}$, $\theta_{23,0}$, and $\theta_{\text{O}_2^*}$
represent the temperature coefficient (dimensionless); $K_{\text{LDOC}}$ represents the Michaelis constant for LDOC (mg
C $L^{-1}$); $K_{\text{Pc}}$ represents the half-saturation constant for phytoplankton limitation (mg C $L^{-1}$); $K_{\text{DO}}$ and $K_{\text{DO}_{\text{O}_2^*}}$
represent the half-saturation constant for DO limitation (mg O $L^{-1}$). More detailed information of these variables
and parameters can be seen in Table A1 and Table A2.

1                                              **(position of Table A1)**

2                                              **(position of Table A2)**

**Appendix B: Sediment flux module**
In this study, a sediment flux module is used to receive the depositional fluxes of particulate organic carbon,
particulate organic nitrogen, and particulate organic phosphorus, which are collectively referred to as particulate
organic matter, from the overlying water. After that, the diagenesis of particulate organic matter will occur in
the sediment and produce soluble end-products. The fluxes of nutrients and SOD across the water-sediment
interface will be determined by the differences in the dissolved concentrations between the resulting sediment
and overlying water combined with the transfer coefficient.
In the sediment flux module, particulate organic matter is classified into three G classes (G1: reactive, G2:
refractory, and G3: inert) with the different reaction rates. The kinetic equation for diagenesis is:

$$H\frac{dG_i}{dt} = -k_{Gi}\theta_{Gi}^{T-20}G_iH + J_{Gi} \qquad (B1)$$

where $H$ is the depth of sediment (m), $G$ represents the particulate organic carbon, the particulate organic
nitrogen, or the particulate organic phosphorus (mg L$^{-1}$), subscript $i$ represents the i$^{th}$ G class (i=1, 2, 3), $k_{Gi}$
represents the corresponding reaction rate (day$^{-1}$), $\theta_{Gi}$ represents the temperature coefficient (dimensionless),
and $J_{Gi}$ represents the depositional fluxes of $Gi$ from the overlying water (g m$^{-2}$ day$^{-1}$).
After the deposition and diagenesis, further reactions of organic matter (including particulate organic
carbon, dissolved organic carbon, particulate organic nitrogen, dissolved organic nitrogen, particulate organic
phosphorus, and dissolved organic phosphorus) will occur in both aerobic layer (denoted as layer 1) and
anaerobic layer (denoted as layer 2). The mass balance equations can be expressed as a general form:

$$H_1\frac{dC_{T1}}{dt} = K_{L01}(f_{d1}C_{T1} - C_{d0}) + w_{12}(f_{p2}C_{T2} - f_{p1}C_{T1}) + K_{L12}(f_{d2}C_{T2} - f_{d1}C_{T1}) - K_1H_1C_{T1} + J_{T1} \quad (B2)$$

$$H_2\frac{dC_{T2}}{dt} = -w_{12}(f_{p2}C_{T2} - f_{p1}C_{T1}) - K_{L12}(f_{d2}C_{T2} - f_{d1}C_{T1}) - K_2H_2C_{T2} - w_2C_{T2} + J_{T2} \qquad (B3)$$

where the subscript 0, 1, 2 represent the overlying water, the aerobic layer, and the anaerobic layer, $H_1$ and $H_2$ represent the thickness of aerobic layer and anaerobic layer (m), respectively, $C_{T1}$ and $C_{T2}$ represent the total concentrations (mg L$^{-1}$) in aerobic layer and anaerobic layer, respectively, $C_{d0}$ represents the dissolved concentrations (mg L$^{-1}$) in the overlying water, $f_{d1}$ and $f_{d2}$ represent the dissolved fractions in aerobic layer and anaerobic layer (dimensionless), respectively, $f_{p1}$ and $f_{p2}$ represent the particulate fractions in aerobic layer and anaerobic layer (dimensionless), respectively, $K_{L01}$ represents the transfer coefficient between the overlying water and aerobic layer (m day$^{-1}$), $K_{L12}$ represents the transfer coefficient between the aerobic layer and anaerobic layer (m day$^{-1}$), $K_1$ and $K_2$ represent the first-order decay rate of in the aerobic layer and anaerobic layer (day$^{-1}$), respectively, $w_{12}$ represents the particle mixing rate between the aerobic layer and anaerobic layer (m day$^{-1}$), $w_2$ represents the sedimentation rate out of the anaerobic layer (m day$^{-1}$), $J_{T1}$ and $J_{T2}$ represent the total influxes for each class of particulate organic matter into the aerobic layer and anaerobic layer (g m$^{-2}$ day$^{-1}$), respectively.

Fluxes of nutrients and DO across the water-sediment interface can be represented as:

$$J = s(C_{water} - C_{sed}) \qquad\qquad (B4)$$

where $s$ represents the transfer coefficient (m day$^{-1}$), $C_{water}$ and $C_{sed}$ represent the concentrations of nutrients and DO in the water and sediment (mg m$^{-3}$ day$^{-1}$), respectively.

**Acknowledge**

This work was supported by the National Natural Science Foundation of China (grant no: 41306105), the Guangdong Natural Science Foundation (grant no: 2014A030313169), the Science and Technology Planning Project of Guangdong Province, China (grant no: 2014A020217003), and the Fundamental Research Funds for the Central Universities (grant no: 17lgzd20).

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

**Table list**
**Table 1.** Overview of model experiments

| Experiments | Description |
| --- | --- |
| **Base** | Forced by the riverine inputs of monthly observed DO, nutrients and particulate organic carbon concentration from 2006 collected by the State Oceanic Administration |
| **DO simulations** | |
| **RivDO-50%** | Same as Base simulation except the riverine DO inputs are decreased by 50% |
| **RivDO+50%** | Same as Base simulation except the riverine DO inputs are increased by 50% |
| **Nutrients simulations** | |
| **RivNtr-50%** | Same as Base simulation except the riverine nutrients inputs are decreased by 50% |
| **RivNtr+50%** | Same as Base simulation except the riverine nutrients inputs are increased by 50% |
| **POC simulations** | |
| **RivPOC-50%** | Same as Base simulation except the riverine inputs of particulate organic carbon are decreased by 50% |
| **RivPOC+50%** | Same as Base simulation except the riverine inputs of particulate organic carbon are increased by 50% |


**Table 2.** Comparisons of nutrient limitation and light limitation on the growth of phytoplankton for Base and two POC simulations. Values are averaged over the bottom layer of the PRE. The lower values represent the stronger limitation.

| | Base | RivPOC-50% | RivPOC+50% |
|---|---|---|---|
| **Nutrient limitation** | 0.81±0.09 | 0.80±0.09 | 0.82±0.09 |
| **Light limitation** | 0.21±0.15 | 0.25±0.16 | 0.18±0.14 |

**Table 3.** A summary of characteristics of hypoxia among three systems (i.e. Chesapeake Bay, northern Gulf of Mexico, and PRE). Abbreviation WCR represents the water column respiration which is the sum of respiration, nitrification, and oxidation.

| | WCR dominant | SOD dominant Autochthonous POC dominant | SOD dominant Allochthonous POC dominant |
|---|---|---|---|
| **Chesapeake Bay** | √ | | |
| **NGOM** | | √ | |
| **PRE** | | | √ |

**Table 4.** A summary of the differences in physical and biogeochemical processes associated with the relative contributions of autochthonous versus allochthonous POC between the PRE and NGOM

| | Period | PRE | NGOM |
|---|---|---|---|
| Allochthonous POC input (t yr$^{-1}$) | Annual | $2.5\times10^{6}$ [a] | $3.8\times10^{6}$ [b] |
| Primary productivity (mg m$^{-2}$ day$^{-1}$) | Summer | 183.9–1,213 [c] | 330-7,010 [d] |
| DIN loading (t d$^{-1}$) | Annual | 1531 [e] | 1955 [e] |
| DIP loading (t d$^{-1}$) | Annual | 27 [e] | 133 [e] |
| DIN:DIP (mol:mol) | Annual | 126 [e] | 33 [e] |
| Residence Time (d) | Summer | 3-5 [f] | ~95 [f] |

[a] Zhang et al. (2013); [b] Wang et al. (2004); [c] Ye et al. (2014); [d] Quigg et al. (2011); [e] Hu and Li (2009);

[f] Rabouille et al. (2008)

**Tabel A1. List of state variables in the water quality model (RCA) and the DO species tracing method**

| variables | Description (unit) |
|---|---|
| DO | Dissolved oxygen (mg $O_2$ $L^{-1}$) |
| $DO_{sat}$ | Saturated DO concentrations (mg $O_2$ $L^{-1}$) |
| $DO_{sed}$ | DO concentrations in the sediment (mg $O_2$ $L^{-1}$) |
| $DO_{BC}$ | DO species which is contributed by lateral boundary condition (mg $O_2$ $L^{-1}$) |
| $DO_{REA}$ | DO species which is contributed by re-aeration (mg $O_2$ $L^{-1}$) |
| $DO_{WCP}$ | DO species which is contributed by water column production (mg $O_2$ $L^{-1}$) |
| $DO_{SOD}$ | DO species which is contributed by sediment oxygen demand (mg $O_2$ $L^{-1}$) |
| $O_2^*$ | Dissolved oxygen equivalent (mg $O_2$ $L^{-1}$) |
| $P_c$ | phytoplankton biomass (mg C $L^{-1}$) |
| RDOC | refractory dissolved organic carbon (mg C $L^{-1}$) |
| LDOC | labile dissolved organic carbon (mg C $L^{-1}$) |
| ReDOC | reactive dissolved organic carbon (mg C $L^{-1}$) |
| ExDOC | algal exudate dissolved organic carbon (mg C $L^{-1}$) |
| $Gi$ | Concentrations of particulate organic carbon, particulate organic nitrogen, or particulate organic phosphorus in i[th] G class (mg $L^{-1}$) |
| $C_{d0}$ | dissolved concentrations in the overlying water (mg $L^{-1}$) |
| $C_{T1}$ | total concentrations in aerobic layer (mg $L^{-1}$) |
| $C_{T2}$ | total concentrations in anaerobic layer (mg $L^{-1}$) |
| $C_{water}$ | concentrations of nutrients and DO in the water (mg $L^{-1}$) |
| $C_{sed}$ | concentrations of nutrients and DO in the sediment (mg $L^{-1}$) |

**Table A2. Main parameters and constants for the water quality model (RCA)**

| parameters | Description (unit) | values |
|---|---|---|
| $\alpha_{OC}$ | oxygen to carbon ratio (mg $O_2$:mg C) | 32/12 |
| $\alpha_{NO_{23}c}$ | oxygen to carbon ratio for nitrate uptake (mg $O_2$:mg C) | 12/14 |
| $\alpha_{ON}$ | oxygen-to-nitrogen ratio (mg $O_2$:mg N) | 32/14 |
| $k_{14,15}$ | nitrification rate at 20 °C (day$^{-1}$) | 0.08 |
| $K_{nitri}$ | half saturation constant for oxygen limitation (mg $O_2$ L$^{-1}$) | 1.0 |
| $k_{20,0}$ | oxidation rates of refractory dissolved organic carbon at 20 °C (day$^{-1}$) | 0.009 |
| $k_{21,0}$ | oxidation rates of labile dissolved organic carbon at 20 °C (day$^{-1}$) | 0.1 |
| $k_{22,0}$ | oxidation rates of reactive dissolved organic carbon at 20 °C (day$^{-1}$) | 0.1 |
| $k_{23,0}$ | oxidation rates of algal exudate dissolved organic carbon at 20 °C (day$^{-1}$) | 0.35 |
| $k_{O_2^*}$ | oxidation rates of dissolved sulfide at 20 °C (day$^{-1}$) | 0.08 |
| $K_{LDOC}$ | Michaelis constant for LDOC (mg C L$^{-1}$) | 0.1 |
| $K_{Pc}$ | half-saturation constant for phytoplankton limitation (mg C L$^{-1}$) | 1.0 |
| $K_{DO}$ | half-saturation constant for DO limitation (mg O L$^{-1}$) | 0.2 |
| $K_{DO_{O_2^*}}$ | half-saturation constant for DO limitation in oxidation of dissolved sulfide (mg O L$^{-1}$) | 0.2 |
| $\theta_a$ | temperature coefficient for re-aeration (dimensionless) | 1.024 |
| $\theta_{14,15}$ | temperature coefficient for nitrification (dimensionless) | 1.045 |
| $\theta_{20,0}$ | the temperature coefficient for oxidation rates of refractory dissolved organic carbon (dimensionless) | 1.08 |
| $\theta_{21,0}$ | the temperature coefficient for oxidation rates of labile dissolved organic carbon (dimensionless) | 1.08 |
| $\theta_{22,0}$ | the temperature coefficient for oxidation rates of reactive dissolved organic carbon (dimensionless) | 1.08 |
| $\theta_{23,0}$ | the temperature coefficient for oxidation rates of algal exudate dissolved organic carbon (dimensionless) | 1.047 |
| $\theta_{O_2^*}$ | the temperature coefficient for oxidation rates of dissolved sulfide (dimensionless) | 1.08 |

**Figure caption**

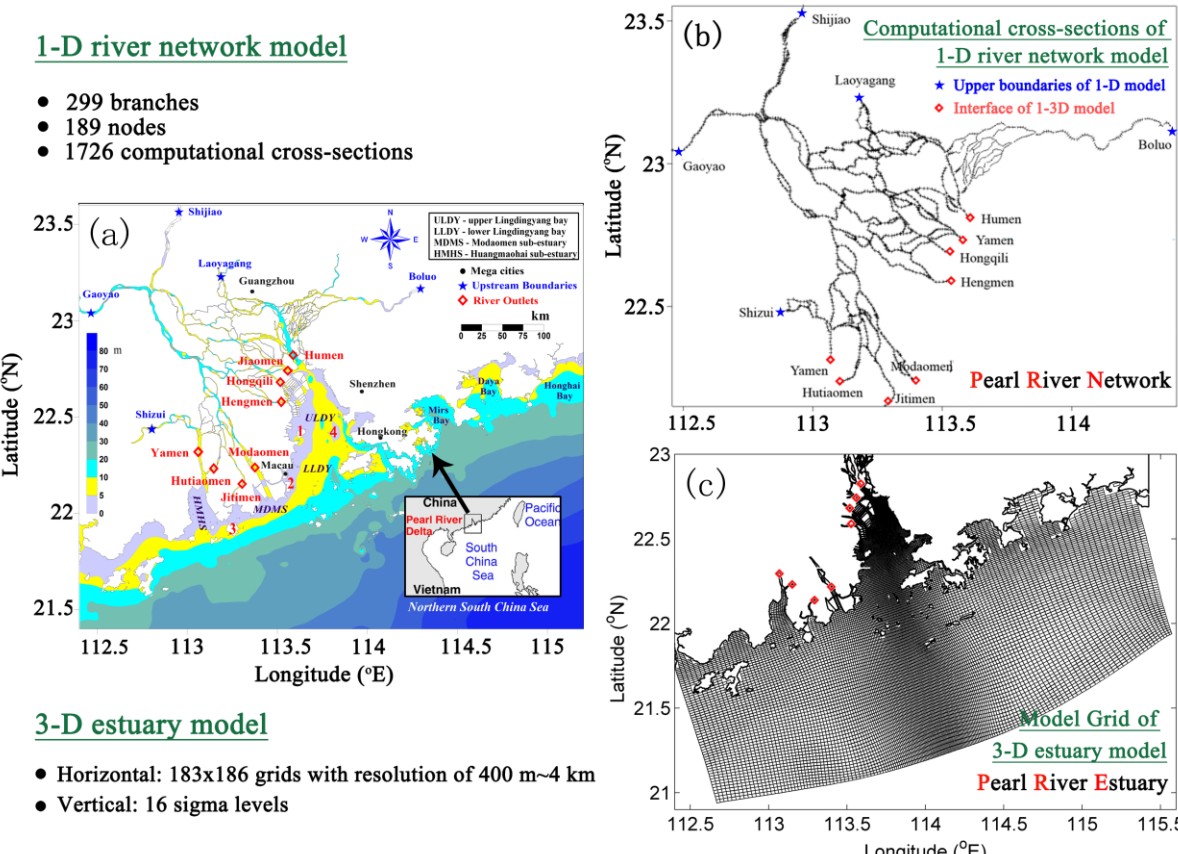

**Figure 1** (a) A bathymetric map showing the Pearl River Network and the Pearl River Estuary, (b)
computational cross-sections for the 1-D river network model, and (c) the model grid for 3-D estuary model.
Red numbers in Figure 1a represent islands which are not marked on the map: 1-Qi'ao island; 2-Hengqin island;
3-Gaolan island; and 4-Inner Lingding island.

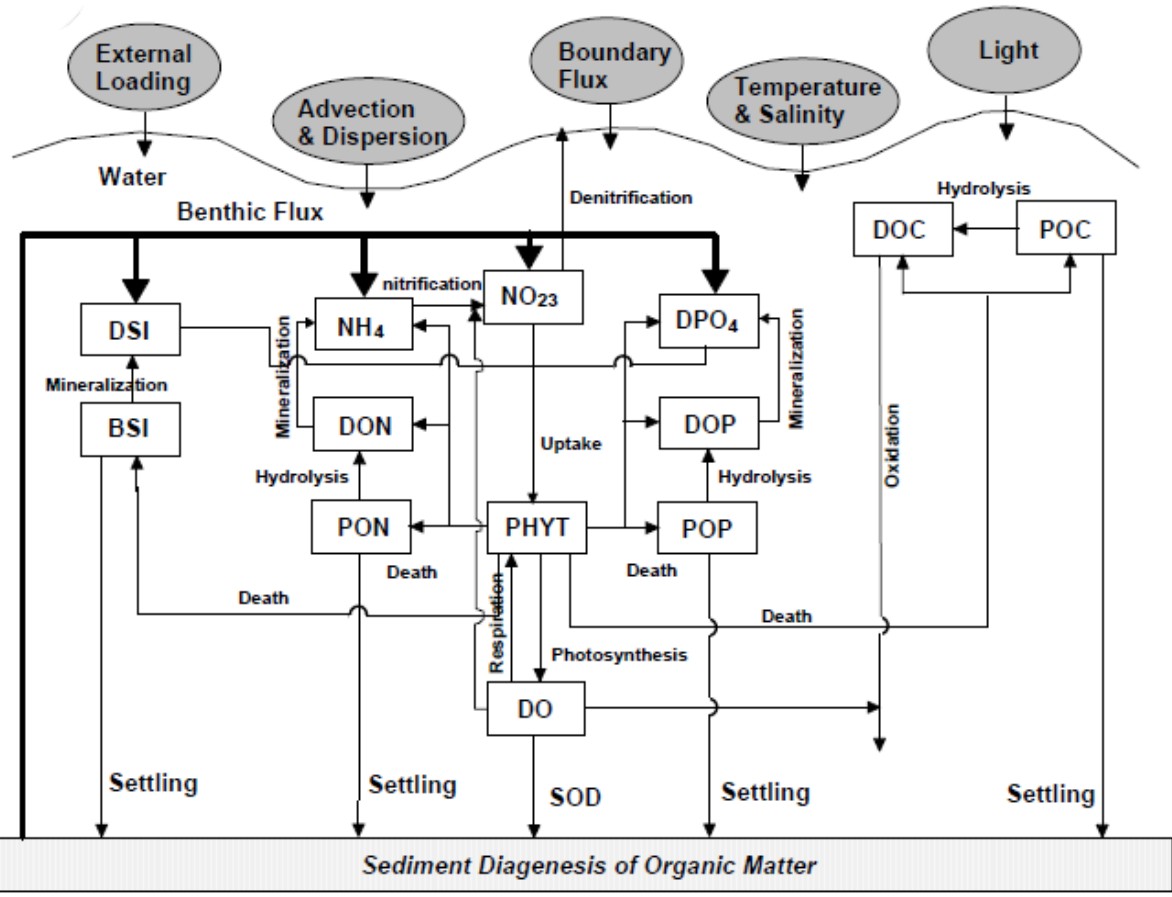

Figure 2 Conceptual framework for RCA model with a sediment flux module (Zhang and Li, 2010). DO represents dissolved oxygen; PHYT represents phytoplankton; POC represents particulate organic carbon; DOC represents dissolved organic carbon; $NH_4$ represents ammonia nitrogen; $NO_{23}$ represents nitrite and nitrate nitrogen; PON represents particulate organic nitrogen; DON represents dissolved organic nitrogen; $DPO_4$ represents dissolved inorganic phosphorus; POP represents particulate organic phosphorus; DOP represents dissolved organic phosphorus; DSi represents dissolved silica; BSi represent biogenic silica; and SOD represents sediment oxygen demand.

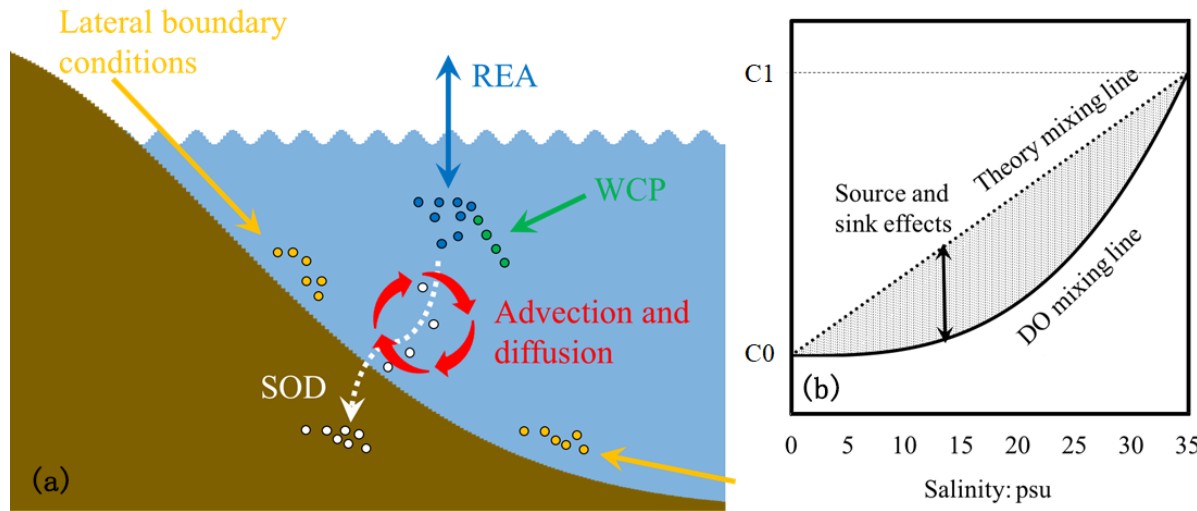

2 **Figure 3** (a) The schematic diagram illustrating the mixing process of dissolved oxygen in the estuary and (b)

3 the schematic plot for dissolved oxygen versus salinity (the solid black curve line) during the mixing in the

4 estuary. C1 represents the concentrations in sea water, while C0 represents the concentrations in river water.

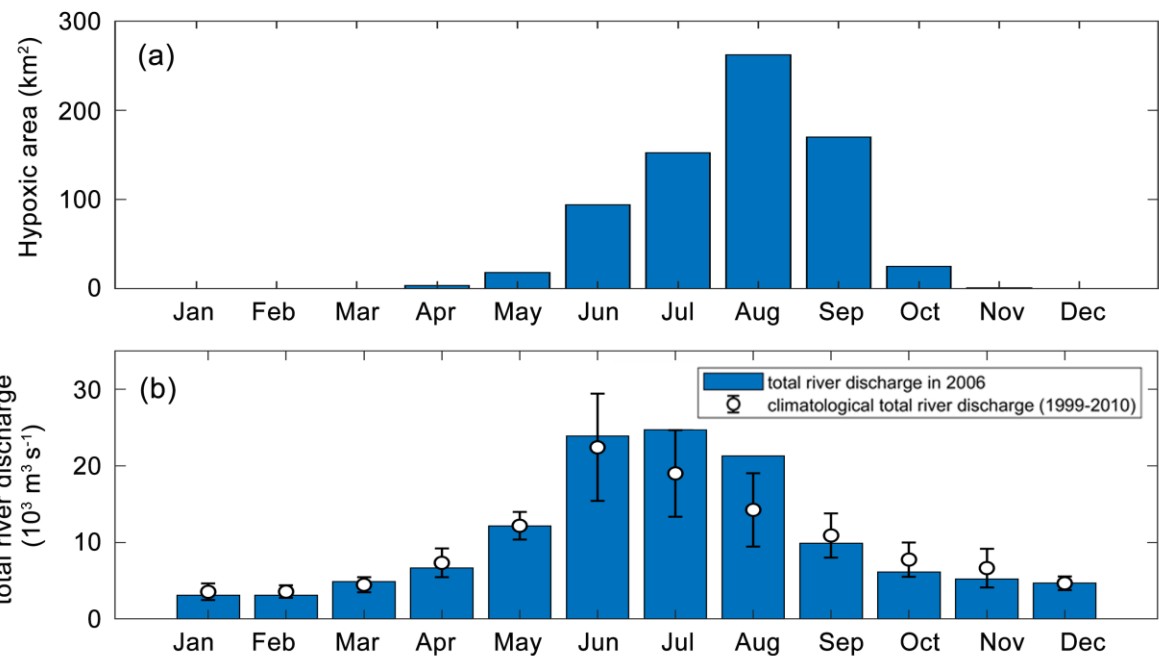

2 **Figure 4**. (a) Annual cycles of the model simulated monthly hypoxic area in 2006 of the PRE, (b) annual cycle

3 of the total river discharges in 2006 (blue bars) and during 1999-2010 (error bars represent a standard deviation

4 around the climatological mean values).

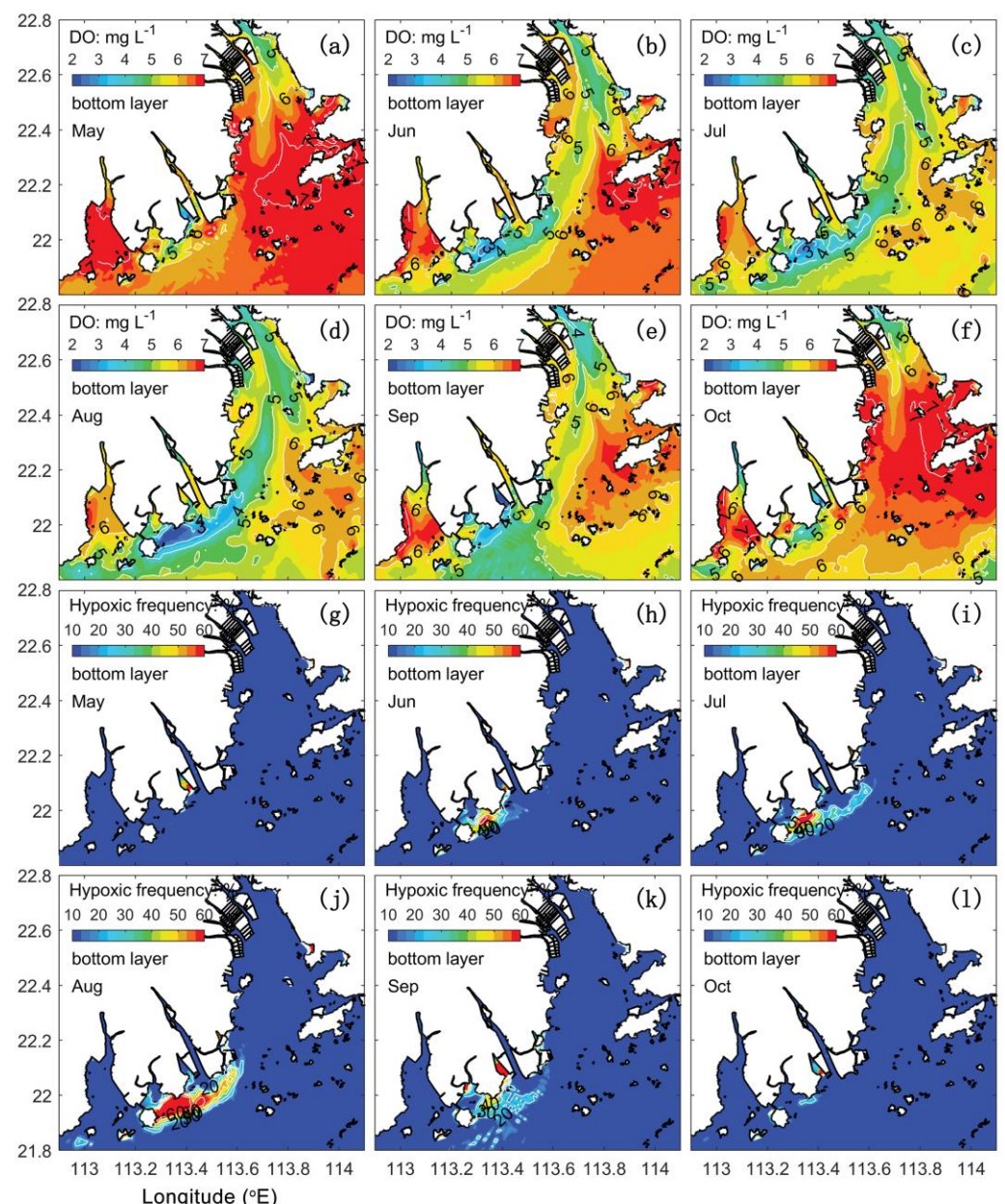

**Figure 5**. Spatial distributions of bottom DO (a~f) and hypoxic frequency (g~l) during the May-October. The

hypoxia is defined as DO concentraion below 3 mg L$^{-1}$.

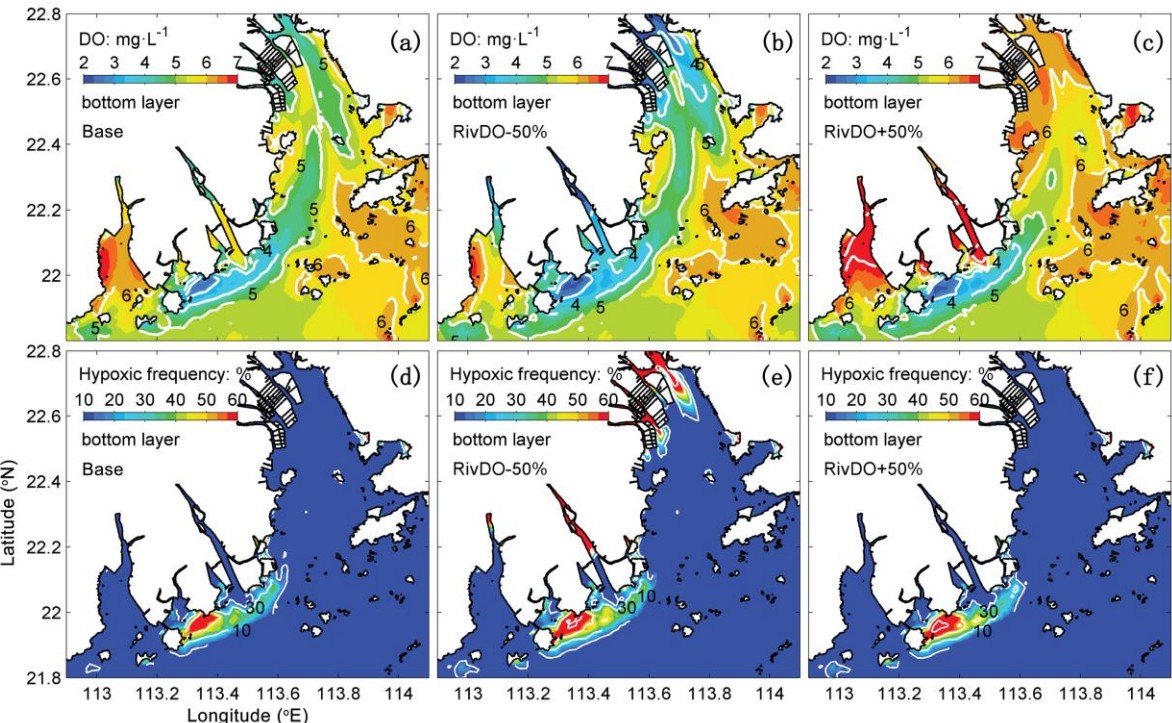

Figure 6 Spatial distributions of DO concentrations (a, b, c) and hypoxic frequency (d, e, f) in the bottom layer

for DO concentration simulations. The DO concentration is averaged over July and August 2006.

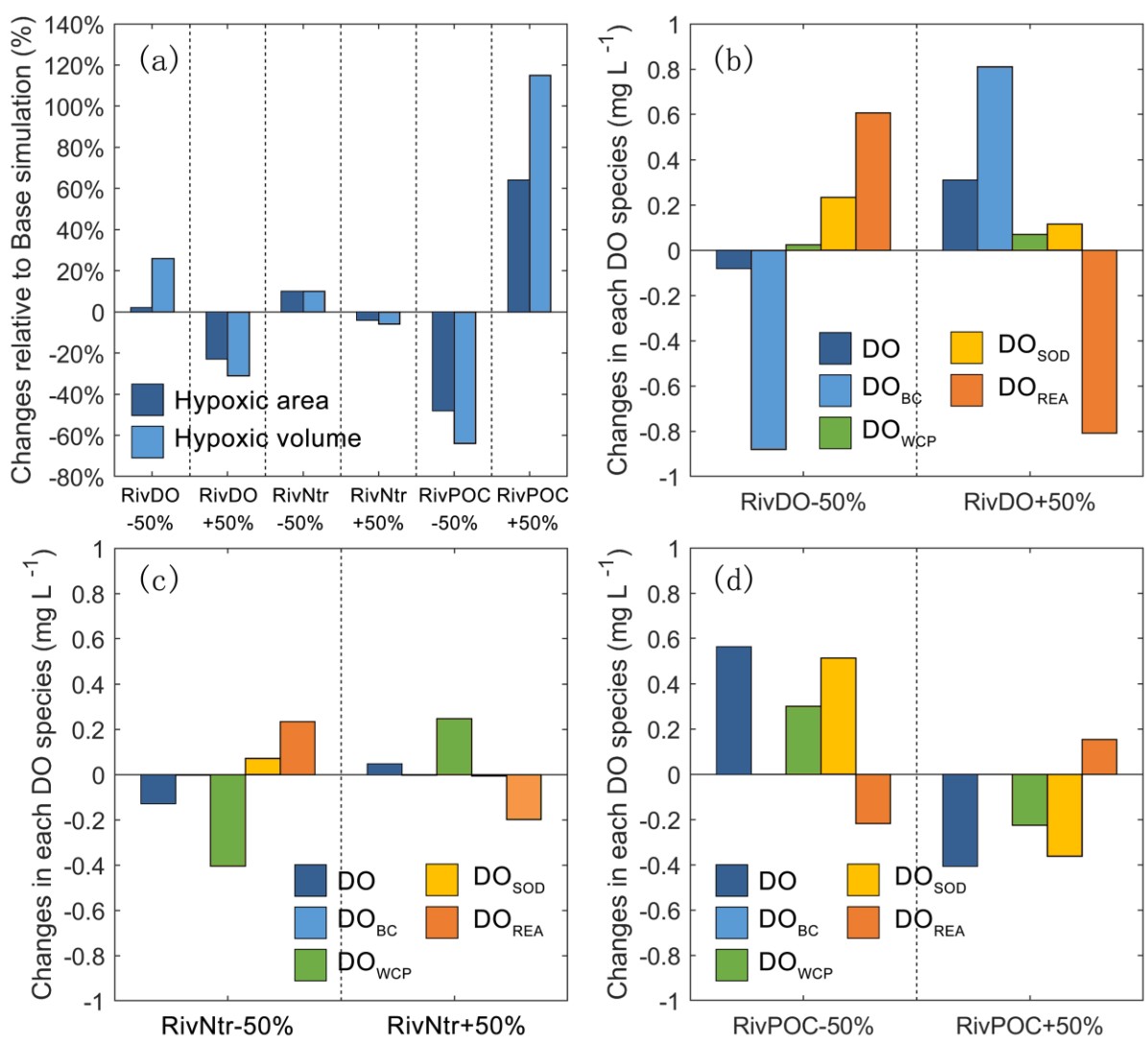

**Figure 7** The percentage changes of the hypoxic area and hypoxic volume in each simulation in relative to the
Base simulation (a). The changes of each DO species averaged over the high hypoxic frequency zone (denoted
as the white contour in Figure 6) in DO simulations (b), nutrient simulations (c), and POC simulations (d) in
relative to the Base simulation.

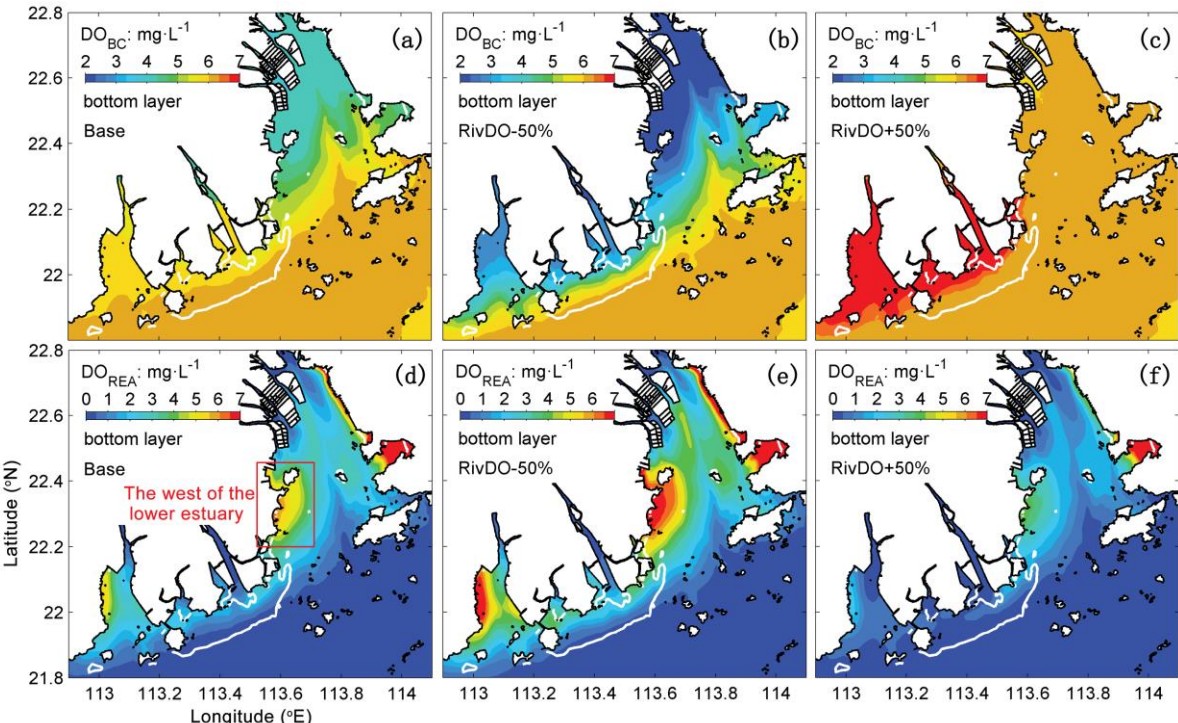

**Figure 8** The spatial distribution of $DO_{BC}$ (a, b, c) and $DO_{REA}$ (d, e, f) concentrations at the bottom layer for three DO simulations. The white contour represents the high frequency zone, and the red box represents the west of the lower estuary.

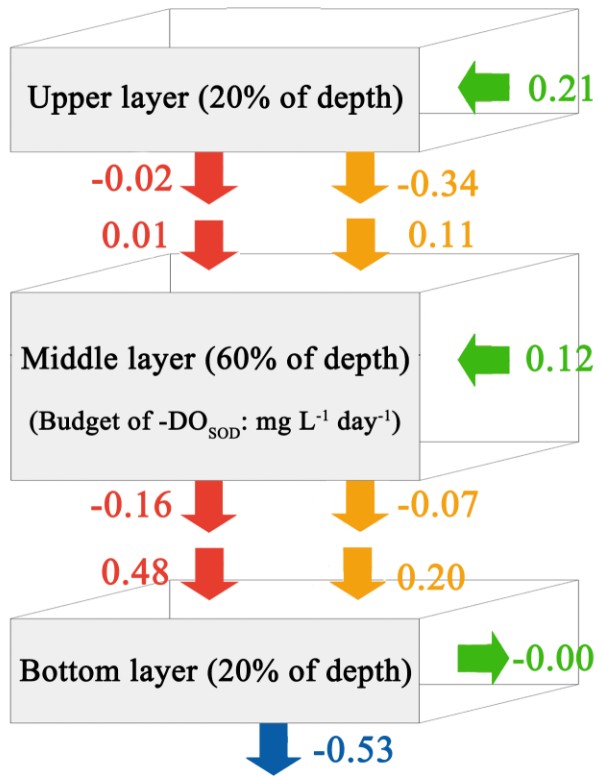

**Figure 9** Budget of $-DO_{SOD}$ for the upper layer, middle layer, and bottom layer in the PRE for the Base simulation. Blue arrows represent sediment oxygen demand, red arrows represent the vertical diffusion, orange arrows represent vertical advection, and green arrows represent horizontal advection. Positive values mean the source effects while the negative values mean the sink effects of the sediment oxygen demand on DO concentrations. (unit: mg $L^{-1}$ $day^{-1}$)

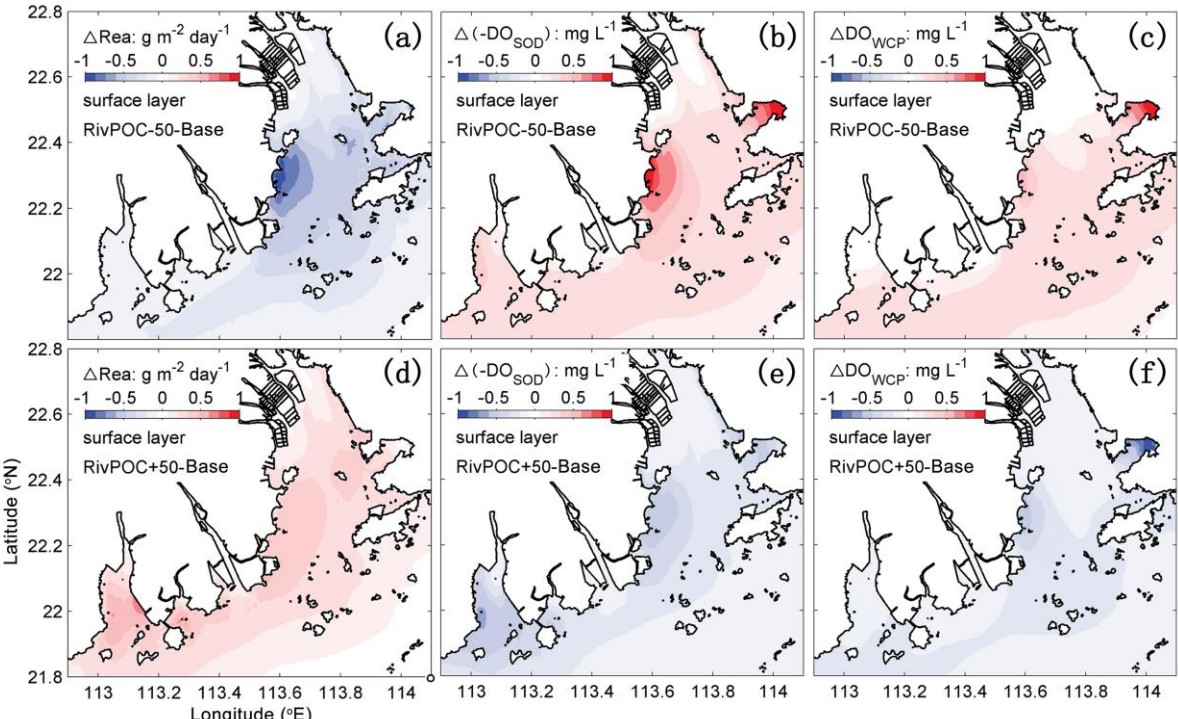

**Figure 10** The changes in air-sea re-aeration rates (a, d), -DO<sub>SOD</sub> (b, e), and DO<sub>WCP</sub> (c, f) concentrations in the surface layer with respect to the Base simulation (model-Base). Positive values of $\Delta Rea$, $\Delta(-DO_{SOD})$ and $\Delta DO_{WCP}$ concentrations represent higher re-aeration rates, higher DO concentrations caused by the changes of the sediment oxygen demand rate and the water column production rate, respectively.

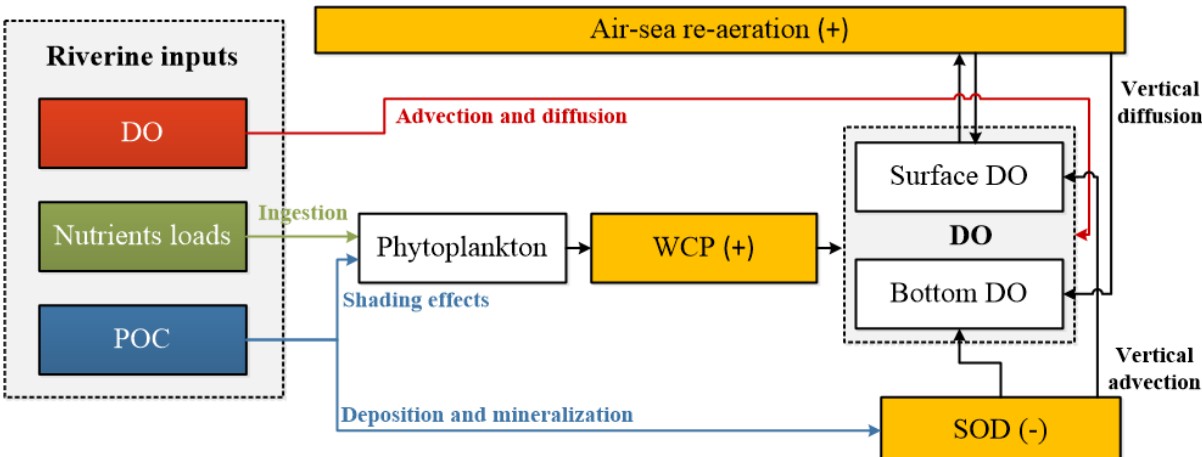

**Figure 11** Conceptual schematic of the oxygen dynamics in response to riverine inputs in the PRE. The white
boxes represent the state variables in the water column, the orange boxes represent the source and sink processes
associated with the oxygen dynamics. The positive signs represent the sources while the negative signs represent
the sinks for DO concentrations.

