# Peer review of "Impacts of anthropogenic inputs on the hypoxia and oxygen dynamics in the Pearl River Estuary"

_Biogeosciences, 2018_

## Referee Comment (RC1) · Anonymous Referee #1 · 12 Jun 2018

**General Comments**

Congratulations for the research work, there is a large amount of work summarised in a clear and well structured document.

**Specific Comments**

[Figure]

1. Page 4, Line 23: The description of the 1-D model does not refer to any other study, Does this mean that the model was developed for this research? Is there any reference for validating this 1-D model?

2. The configuration description of the 1-D model is quite small in contrast with the ones for the 3-D model and the water quality model.

3. Page 7, Lines 13 and 14: mention any quantitative description for the temperature and salinity validation as it is done for the tide.

4. Model Validation section: No validation mentioned of the 1-D model

5. Section 2.2: Are there any other hypoxia events reproduced by the model throughout the period Nov. 2005 to Dec. 2006, apart of the summer 2006? Are these events observed or not observed? Is the hypoxia event of summer 2006 the only event simulated by the model?

6. A map showing the location of the study area in a global context will be a great help for the reader which is not familiar with the study area.

**Technical Corrections**

- Page 2, Line 17: Why the reference is made on Italic font (Diaz and Rosenberg)

- Not sure which the format for the references within brackets:
  Page 2 Lines 19 and 20
  Page 5 Line 19
  Page 6 Line 8

- Page 10, Line 8: where is the definition for **Cont**?
  I see there are the names for each simulations. Could be possible to mention this

[Figure]

before start describing each of them? (The 7 simulations are named as . . . . . . . . and summarized in table 1)

- Figure 1a [page 24]: It could be just the pdf copy, but the y-axis (latitude) top and bottom labels are missing a 2 (23.5 and 21.5)

**Suggestions**

- Page 2, Line 13: Enhance instead of exaggerate

- Page 5, Line 8: There is a reference for the Mellor-Yamada model but not for the Smagorinsky-type formula. It should be a reference for each of them as they are in the same sentence (line).

- Page 7 Line 7: Could be possible to specify if the summer is on the north or south hemisphere? It is in the north hemisphere but the suggestion points to be specific as the moths to consider are not the same ones.

- Page 18, Line 4: anthropogenic perturbations instead of just perturbations

---

## Referee Comment (RC2) · Anonymous Referee #2 · 27 Jun 2018

Review of "Impacts of anthropogenic inputs on the hypoxia and oxygen dynamics in the Pearl River Estuary" by Wang, Hu, Li, Yu and Huang

Summary:

In this manuscript the authors use a physical-biogeochemical model to examine a hypoxic event in the Pearl River Estuary (PRE) in July and August 2006. They conduct several numerical experiments in order to determine the relative impact of riverine inputs of oxygen, nutrients and organic matter on hypoxia in the PRE. They specifically examine three processes that affect oxygen dynamics: re-aeration due to air-sea oxygen flux, sediment oxygen demand, and all remaining processes which together is

referred to as WCP (water column production). This is a well-written manuscript with some very interesting results, but some clarifications, some more discussion, and a few additional experiments should be performed before publication. The comments below are lengthy, but if addressed fully the resulting paper would be a very valuable contribution to Biogeosciences.

Major comments:

As I understand it, all results shown in the manuscript are for July and August 2006. This should be made clearer in the abstract, which is written more like this is the "general" case for the PRE. I understand that the model has only been evaluated for July and August 2006, so we don't really know whether the oxygen concentrations at other times of the year are correct or not; however, as a reader I was very interested to see results for the whole summer (May to September), or even for the whole year, rather than just for two months of one year. How does the temporal variability of hypoxia change in the numerical experiments? This analysis does not seem complete without this addition.

As a reader, I was also wondering whether July and August 2006 was a typical year. Was 2006 a particularly dry July/August? Or wet time period? Are the results of the sensitivity experiments conducted here likely to hold in other years?

One of the main results of this manuscript was that hypoxia in the PRE is not sensitive to nutrient concentrations of the river water entering the region (unlike the Chesapeake Bay and the Gulf of Mexico, for example). This result, however, has to be at least slightly dependent on what value is used for the nutrient concentrations in the eight rivers. What concentrations are used and are they realistic? Where do these concentrations come from? A terrestrial-biogeochemical or watershed model? More detail is needed here. Also, it sounds as if only the nutrient concentrations were changed in the largest river, not all eight rivers. The authors need to show results of changing the concentrations in all eight rivers, not just the largest, since the smaller ones closest to the

hypoxic zone might impact the hypoxia zone more than the large river, which is farther from the region of hypoxia. (The same is true for the oxygen and POC experiments.)

This analysis compares the impact of sediment oxygen demand, re-aeration and WCP on hypoxia. However, this is misleading since WCP is the sum of multiple positive and negative terms. Thus this term is likely smaller than its components. For a more complete analysis, the authors need to separate out the various components of WCP, including respiration, nitrification, water column remineralization etc... This is particularly important because in the discussion they state that in the PRE water column respiration/remineralization is not as important as it is in places such as the Chesapeake Bay. But we cannot see this (truly interesting!) result unless the authors isolate these terms.

It is not completely clear why the "physical modulation" method is needed. If this is a fully coupled physical-biogeochemical model (as is stated), then why can't the authors simply save each of the oxygen flux terms in the oxygen budget? Presenting results in units of DO per unit time (as is done in Figure 7) would be much more helpful for the reader. The idea of different "species" of oxygen seems a bit convoluted. Clearly REA, WCP and SOD have units of oxygen per unit time (see equation 1). Showing figures of these quantities, rather than DO_REA, DO_WCP and DO_SOD would make the manuscript more clearly understandable to readers.

I really like the idea that re-aerated surface waters can penetrate to the bottom water and offset the changes in DO caused, for example, by increased nutrient, DO, or OM riverine inputs. The authors discuss that this is not the case on the Gulf of Mexico shelf, where hypoxia occurs as a very thin layer near the bottom. The comparison and emphasis on the Gulf of Mexico seems a bit out of place, since the PRE seems to be more similar to the Chesapeake Bay in many ways. The discussion could be strengthened by making a three way comparison between the Chesapeake, Gulf of Mexico and the PRE. Isn't the re-aeration process described here similarly important in the Chesapeake Bay, where hypoxia occurs as a thick layer, which is not far from the

surface in a typical July/August?

There is some considerable overlap with the authors' previous publication (Wang et al., 2017, BG). For instance, it appears to me that one of the main points in the abstract of the current manuscript: "Model results showed that hypoxia in the Pearl River Estuary was confined to the shelf off the Modaomen sub-estuary with a hypoxic area of ~200 km2 mainly due to the combined effect of re-aeration and sediment oxygen demand" was actually a primary result of this previous publication. This should be made clearer in the abstract and introduction. Clearly this study builds off the previous study. Although the previous study is mentioned in the abstract, the differences between the current study and the previous study should be made clearer to the readers.

Minor Comments:

Abstract last sentence – suggest changing this to: "This study highlights the importance of re-aeration in determining the hypoxic extent and in reducing hypoxia variability in shallow estuaries."

Abstract – Define here (and in introduction) that by re-aeration you mean a flux of oxygen across the air-sea interface. (Currently this doesn't occur until page 6).

Introduction – Authors could mention climate change as another anthropogenic impact, since recent studies are showing that increasing temperatures have a large impact on increasing hypoxia.

P2, line 7: Why is there a ten-year lag? Does this occur in an estuary like the PRE? Or maybe it's not relevant here.

Page 4, line 11: This paragraph is talking about how nutrient inputs to the Pearl River Estuary can impact hypoxia, but this line is about particulate organic carbon, which could be moved to the following paragraph talking about organic matter.

Page 4, line 16: What is the organic matter? Is it only POC mentioned in line 11? Or does it include PON (nitrogen) and dissolved organic matter? Which type of organic

matter primarily contributes to hypoxia?

P4, line 20: How are these models dynamically coupled? If these were dynamically coupled, the estuarine model would provide feedbacks to the riverine model. Is that the case? Also, the model set up seems to assume that there are no freshwater or nutrient sources (from the land) into Mirs Bay, Daya Bay or Honghai Bay. Is there evidence to support this assumption?

P5, "Water quality model" section: In this section the authors need to describe more clearly where their riverine biogeochemical concentrations are derived from, since these are at the very heart of their numerical experiments. Do concentrations of the 26 state variables all come from the riverine model described above? If so, more information regarding the details of the biogeochemistry of the riverine model is needed. Where do the outer boundary conditions come from, for the estuarine model? How about atmospheric deposition of nutrients, like nitrate and phosphate? Are all these assumed to be negligible? How realistic is this assumption?

P6, line 1: Since one of the conclusions of the manuscript is the relative importance of SOD compared to WCP (see abstract), here the terms making up "WCP" need to be written out explicitly.

P6, line 7: Please provide the equations for photosynthesis, respiration, nitrification and oxidation (potentially in an appendix), and provide values of all parameters used. (The reference used here for the model is a white paper from 14 years ago. The model has been adjusted since then. Are the authors really using those original parameters and equations? Please include information on the version of the model that is being implemented.)

P6, line 12: Define what is meant by "dissolved matter". Is this dissolved organic matter, i.e. DON and DOC? Or dissolved nutrients, i.e. ammonium? Or both?

P6, line 18: As above, please provide values of these parameters within this paper

(possibly in an appendix.)

P6, line 27: As above, please provide equations and parameter values for DO_sed (possibly in an appendix)

P7, line 2: Earlier the authors stated that this is a dynamically coupled model, but here it sounds as if the water quality model is being run offline from the physical model, which would indicate that there is no dynamic coupling, and the biological simulation cannot impact the physics. Please make it clearer in the text as to whether the models are truly dynamically coupled, or simply run offline.

P 7 line 4: What data is being referre to here and how was it used? Data assimilation? Forcing? Validation?

P7, line 7: There are actually very few observations of DO presented in Wang et al. (2017). Are these really the only observations available of DO in the PRE region? Is nothing more available since 2006? It also looks like the oxygen data shown in Wang et al. rarely, if ever, actually go hypoxic?

P8, line 3: Because the authors have only evaluated model results for oxygen in July and August 2006, does this mean these results only are valid for that year? Is that a particularly wet year or a dry year? Or an average year? Can you put this year in perspective? (Perhaps in the discussion?)

Page 8, line 13: This sentence seems to indicate that this estimation is not straightforward only in river dominant estuaries. How about tide dominant estuaries, which can also be impacted by local and remote source and sink processes?

P9, line 8: What does "Cont" stand for? Continuous? I would think "Base" or "Reference" or "Realistic" might be better descriptions of this simulation.

Section 2.3: The text is not clear here. Are the concentrations of DO and nutrients reduced in all 8 rivers, or only the Humen? Also it is not clear whether the concentrations of DO and nutrients in the experiments are set to what is predicted in 2050, or are

simply increased by 50%. In reality, the concentrations in 2050 will depend on management decisions which are very difficult to predict. I think it's best to state here that you increased/decreased the concentrations by 50%, and if you want to convince the reader that these are representative of 2050 and 1970 respectively, then bring this up in the discussion. Please provide the concentrations of DO and nitrate (as an example nutrient) used in each of these experiments. More detail is needed here. If freshwater flows stay the same, this should be stated.

P11, line 19: Remove HFZ acronym since it is not used elsewhere. Please define the hypoxic frequency zone more quantitatively since this is used throughout the text. Where exactly is this? It's hard for the reader to know. Does it change in time?

P11, line 23: The word "additionally" should come before "occurs" since hypoxia also occurs on the shelf.

P13, line 9: Also list percent changes in hypoxia area and volume, as was done above.

P13, line 22: Considering using PRE acronym earlier. (It hasn't been used much since very early in the manuscript.)

P14, line 14: Aren't there two POC simulations/experiments, not three?

P14, line 24: In the results, it would make sense to discuss Figure 7 (the "Cont" results) before the sensitivity experiment results, rather than inside the section 2.3 sensitivity experiment section.

P14, line 25: The figure shows 0.53, not 0.55?

P15, line 1: How does the reader compute 0.13 from Figure 7?

P15, line 2: Based on equation 8, I would think the dark blue DO bar would equal the sum of all the other bars, but this doesn't seem to be the case? Why is this?

P15, line 11: It is important to qualify the 217km2 statistic by saying that this is true only for a July/August average in 2006. This is not true for other months of the year,

and we don't know whether this is true for other years.

P15, line 15: DO_REA is not a term that your readers will be familiar with (unless they have read this paper carefully). This paper will have a greater impact if this could be reworded such that processes are mentioned, i.e. discuss the re-aeration of surface water via air-sea flux (in units of oxygen per unit time), rather than DO_REA.

P15, line 19: Again where is the hypoxic frequency zone? Where is "the west of the lower estuary"? Also, make it clearer that this is a result of Wang et al. (2017) and not of this paper.

P15, lines 4-8: This is a very interesting result! But unfortunately this paper does not show any statistics on water column respiration, so this is not clear. Please separate out the various terms inside WCP so the reader can see specifically that water column respiration is not large here.

P16, line 15: I don't think the authors mean the residence time of the Mississippi River, which extends a great distance, well up into the continent of North America. Do you mean the shelf plume area? This section would be much stronger if the authors compared all three systems mentioned here: the GoM, Chesapeake Bay and the PRE.

P16: Rather than discussing terrestrial vs. marine POC, I think it would be clearer to discuss autocthonous vs. allochthonous POC. "Marine POC" sounds as if it comes from outside the hypoxic zone from the ocean, but I don't think this is what is meant?

P16, line 22: Is July-August a wet or dry season?

Section 4.1: This section needs to describe more completely the difference in marine vs. terrestrial POC in the PRE vs. Gulf of Mexico. Why does terrestrial POC not impact hypoxia? Just because that the POC entering from the river is relatively small and sinks out before making it al the way to the shelf? Or is there something specifically different about the terrestrial matter entering from the Mississippi compare to that being delivered to the PRE? Is this a residence time issue? Is the terrestrial source more

important in the PRE because the nutrient inputs are quite low, compared to what they are in the Gulf of Mexico? What about in the Chesapeake Bay?

P17, line 17: Isn't the same likely to occur in Chesapeake Bay? This might be a very interesting discussion point here.

Figures:

Figure 1: The figures do not look to be italicized (as it says in the caption). Fig 1b does not add any significant information to what appears in Fig 1a. In Fig 1c "grids" should be "grid". The Fig 1a caption should note that this is a bathymetric map.

Figure 3: The text refers to 3a and 3b, but the left panel is not marked (b), and there is no reference to (b) in the caption.

Figure 5: The y-axes in (b)-(d) should specify that these are "Changes in concentration", not concentrations themselves. Also, (b)-(d) should have same y-range to make it easier for the reader to compare all three figures.

Figure 6: Please label figures (a)-(e) and provide captions for each. What is the white line? Axes are not labeled.

Figure 7: This figure is a little confusing, because one would expect that the vertical diffusion out of box 1 would represent the vertical diffusion into box 2. I gather the net diffusion arrows are shown, but maybe it would make more sense to show the middle layer as having a +0.02 diffusion into the middle layer at the top, and a -0.17 diffusion out of the middle layer at the bottom? But why doesn't this equal 0.48? Maybe I'm confused because this is only DO_SOD, and not total oxygen? Wouldn't this be a more enlightening figure if all the DO fluxes were shown here?

English language comments:

Throughout, "organic matters" should be changed to "organic matter". And similarly "dissolved matters" should be "dissolved matter".

P 4, line12 – processes should be process

P6, Line 26 – transportation should be transport

P7, line 7 – delete "here" and "as"

P8, line 16 – should be "interacting"

Page 12, line 1: "further" should be "farther"

P14, line 17: should be " layer, exerting a strong constraint"

P15, line 14: supply should be supplies

P15, line 24: most should be "more"

P16, line 5: should be "are the most important processes"
* * *

---

## Author Comment (AC1) · 1 Aug 2018

We thank the reviewer for the constructive comments and suggestions that are very helpful to the revision of our manuscript.

Detailed response to all comments are given below (responses are shown in blue)

**Anonymous Referee #1**

**General Comments**
Congratulations for the research work, there is a large amount of work summarised in a clear and well structured document.

**Specific Comments**

1. Page 4, Line 23: The description of the 1-D model does not refer to any other study, Does this mean that the model was developed for this research? Is there any reference for validating this 1-D model?
**Response:**
    The 1-D model was configured and coupled with the 3-D model as detailed in Hu and Li (2009). The 1D-3D coupled model has been validated and applied to study the water-nutrients-sediment budgets (Hu and Li, 2009; Hu et al., 2011), the oxygen budget (Wang et al., 2017), and the nutrient fluxes between the sediment and overlying waters in the Pearl River Estuary (Liu et al., 2016) . In the revised manuscript, we will include more details of the 1D-3D coupled model to make it clearer. References for the validation are given below.

2. The configuration description of the 1-D model is quite small in contrast with the ones for the 3-D model and the water quality model.
**Response:**
    As suggested, we will include more descriptions of the 1-D model in our revised manuscript.

3. Page 7, Lines 13 and 14: mention any quantitative description for the temperature and salinity validation as it is done for the tide.
**Response:**
    As suggested, we will include some quantitative description for the temperature and salinity validation in our revised manuscript. More detailed validations can be referred to the Section 3 in our previous study (Wang et al., 2017).

4. Model Validation section: No validation mentioned of the 1-D model
**Response:**
    In this study, we use the 1D-3D coupled model with a purpose to account for the interactions of hydrodynamics between the river network and the estuary. The 1-D and the 3-D model were run in parallel and they exchange model quantities across the coupling interface. The eight outlets (shown in Figure 1 in original manuscript) are the exchange interface of the 1-D and 3-D models, which serve as the lower boundaries of the 1-D model and at the same time the upper boundaries of the 3-D model. At each time step, the 3-D model utilizes the simulated discharges obtained from the 1-D model as the river boundary forcing, and sends the simulated water levels to the 1-D model as the downstream boundary forcing as a feedback. Therefore the eight outlets are very important for the assessment of the coupled model performance. We have validated the simulated water

levels and/or river discharges against observations at eight outlets in years 1999 (Hu and Li, 2009) and 2006 (Wang et al., 2017 and P7 line 8-11 in the our manuscript). Note that the validations at eight outlets are for both 1-D and 3-D models. In our revised manuscript, we will provide more details of the 1D-3D coupled model-configuration and validations to make it clearer.

5. Section 2.2: Are there any other hypoxia events reproduced by the model throughout the period Nov. 2005 to Dec. 2006, apart of the summer 2006? Are these events observed or not observed? Is the hypoxia event of summer 2006 the only event simulated by the model?

**Response:**

As shown in Figure r1, our model simulates hypoxia from April to October in 2006. In the simulation, the hypoxia starts to develop in April, peaks in August, and disappears in October. However, oxygen observations are only available in July and August 2006 when hypoxia are observed, while for other months no observations are available for validating the model simulated hypoxia. This is one of the main reasons why in the manuscript we only focused on July and August in 2006 to study the impacts of riverine inputs on hypoxia and oxygen dynamics in the Pearl River Estuary. Additionally, previous reported hypoxia also mainly occurred in July and August (Cai et al., 2013). Plus, from the aspect of the river discharges from the Pearl River network, these two months are typical wet seasons with the monthly-averaged total river discharges over 20,000 $m^3$ $s^{-1}$ and the year 2006 is a wet year with the annual averaged total river discharges exceeding 10,000 $m^3$ $s^{-1}$ (Figure r2).

We agree that it would be an interesting topic to study the annual cycle and multi-years variations of hypoxia in the Pearl River Estuary. However, it will be quite hard to study the annual cycle and multi-years variations of hypoxia in this region due to the insufficiency of observational data. And to our knowledge, there are currently few studies on these two topics. Nevertheless, we believe that our study can provide some scientific basis and guidance for further modelling or observational studies on the hypoxia in the Pearl River Estuary.

[Figure]

Figure r1. Distributions of the monthly averaged bottom DO and annual cycle of the hypoxic area in the Pearl River Estuary

**Monthly discharge in 2006**

[Figure]

Figure r2. The annual cycle of monthly-averaged total river discharges in 2006

6. A map showing the location of the study area in a global context will be a great help for the reader which is not familiar with the study area.

**Response:**

We tried as suggested but found it hard to find the Pearl River Estuary in a global map. Alternatively, we will show the location of the Pearl River Estuary in the map of South China Sea in the revised manuscript. The revised figure is shown below.

Figure r3. Maps showing (a) the Pearl River Delta with the Pearl River network and the Pearl River Estuary, (b) computational cross-sections for 1-D river network model, and (c) model grid for 3-D estuary model.

**Technical Corrections**
• Page 2, Line 17: Why the reference is made on Italic font (Diaz and Rosenberg)

• Not sure which the format for the references within brackets:
Page 2 Lines 19 and 20
Page 5 Line 19
Page 6 Line 8
**Response:**
    We will double check and correct the format of references throughout the manuscript.

• Page 10, Line 8: where is the definition for **Cont**? I see there are the names for each simulations. Could be possible to mention this before start describing each of them? (The 7 simulations are named as . . . and summarized in table 1)
**Response:**
    As suggested, we will define the name for each simulation before describing them in the revised manuscript.

• Figure 1a [page 24]: It could be just the pdf copy, but the y-axis (latitude) top and bottom labels are missing a 2 (23.5 and 21.5)
**Response:**
    We have revised the figure, please see Figure r3..

**Suggestions**
• Page 2, Line 13: Enhance instead of exaggerate
**Response**:
    Revised as suggested.

• Page 5, Line 8: There is a reference for the Mellor-Yamada model but not for the Smagorinsky-type formula. It should be a reference for each of them as they are in the same sentence (line).
**Response**:
    We will add the reference in our revised manuscript.

• Page 7 Line 7: Could be possible to specify if the summer is on the north or south hemisphere? It is in the north hemisphere but the suggestion points to be specific as the moths to consider are not the same ones.
**Response**:
    We will modify the sentence in the revised manuscript as 'The coupled physical-biogeochemical model has been validated against available observations in July and August 2006 in Wang et al. (2017)'.

• Page 18, Line 4: anthropogenic perturbations instead of just perturbations
**Response**:
    As suggested, we will use the term 'anthropogenic perturbations' in the revised manuscript

**Reference**

Hu, J. and Li, S.: Modeling the mass fluxes and transformations of nutrients in the Pearl River Delta, China, J. Mar. Syst., 78(1), 146–167, doi:10.1016/j.jmarsys.2009.05.001, 2009.

Hu, J., Li, S. and Geng, B.: Modeling the mass flux budgets of water and suspended sediments for the river network and estuary in the Pearl River Delta, China, J. Mar. Syst., 88(2), 252–266, doi:10.1016/j.jmarsys.2011.05.002, 2011.

Liu, D., Hu, J., Li, S. and Huang, J.: Validation and application of a three-dimensional coupled water quality and sediment model of the Pearl River Estuary, Huanjing Kexue Xuebao/Acta Sci. Circumstantiae, 36(11), 4025–4036, doi:10.13671/j.hjkxxb.2016.0145, 2016 (in Chinese with English abstract).

Wang, B., Hu, J., Li, S. and Liu, D.: A numerical analysis of biogeochemical controls with physical modulation on hypoxia during summer in the Pearl River estuary, Biogeosciences, 14(12), 2979–2999, doi:10.5194/bg-14-2979-2017, 2017.

---

## Author Comment (AC2) · 1 Aug 2018

We thank the reviewer for the constructive comments and suggestions that are very helpful to the revision of our manuscript.

Detailed response to all comments are given below (responses are shown in blue)

**Summary:**

In this manuscript the authors use a physical-biogeochemical model to examine a hypoxic event in the Pearl River Estuary (PRE) in July and August 2006. They conduct several numerical experiments in order to determine the relative impact of riverine inputs of oxygen, nutrients and organic matter on hypoxia in the PRE. They specifically examine three processes that affect oxygen dynamics: re-aeration due to air-sea oxygen flux, sediment oxygen demand, and all remaining processes which together is referred to as WCP (water column production). This is a well-written manuscript with some very interesting results, but some clarifications, some more discussion, and a few additional experiments should be performed before publication. The comments below are lengthy, but if addressed fully the resulting paper would be a very valuable contribution to Biogeosciences.

**Major comments:**

1. As I understand it, all results shown in the manuscript are for July and August 2006. This should be made clearer in the abstract, which is written more like this is the "general" case for the PRE. I understand that the model has only been evaluated for July and August 2006, so we don't really know whether the oxygen concentrations at other times of the year are correct or not; however, as a reader I was very interested to see results for the whole summer (May to September), or even for the whole year, rather than just for two months of one year. How does the temporal variability of hypoxia change in the numerical experiments? This analysis does not seem complete without this addition.

**Response:**

We will make it clearer in the revised manuscript that the results are based on July and August 2006 only. As suggested, we will show results of hypoxia for the whole summer (May to September) in our revised manuscript. Plus, we will also provide more justifications on why we focused on July-August only and add discussions on how representative or comparable is the hypoxia in this time of the year to that of other months or years.

In terms of validations, in this manuscript we have only presented the model-data comparison results in July and August 2006. However, the coupled physical-biogeochemical model has been thoroughly validated against not only physical variables (i.e. water levels, salinity, and temperature), DO concentrations, but also the historical observations of some important biological variables (e.g. chlorophyll and particulate organic carbon) and processes (i.e. re-aeration, sediment oxygen demand, and primary productivity) (see Wang et al., 2017). Our model is able to reproduce the observed hypoxia near the Modaomen sub-estuary and the main processes associated with DO. Previous studies have also reported the hypoxia near the Modaomen sub-estuary with the similar spatial extents and characteristics as the model simulated (Cai et al., 2013; Lin et al., 2001; Zhang and Li, 2010).

With regard to the temporal variability of model simulated hypoxia, Figure r1 shows that the simulated hypoxia in 2006 starts to develop in April, peaks in August, and disappears in October. However, oxygen observations are only available in July and August 2006 when hypoxia are

observed, while for other months no observations are available for validating the model simulated hypoxia. This is one of the main reasons why in the manuscript we only focused on July and August in 2006 to study the impacts of riverine inputs on hypoxia and oxygen dynamics in the Pearl River Estuary. Additionally, previous reported hypoxia also mainly occurred in July and August (Cai et al., 2013). Plus, from the aspect of the river discharges from the Pearl River network, these two months are typical wet seasons with the monthly-averaged total river discharges over 20,000 m3 s-1 and the year 2006 is a wet year with the annual averaged total river discharges exceeding10,000 m3 s-1 (Figure r2).

---

## Author Response (AR1)

We thank the editor and reviewer for the constructive comments and suggestions that are very helpful to the revision of our manuscript.

Detailed response to all comments are given below (responses are shown in blue). A revised manuscript with changes marked blue is submitted along with the response.

**Editor**

**Specific Comments**

1. R2. "As shown in Figure r2, the year 2006 is a wet year with the total river discharge over 10,000 m3 s-1 and the monthly river discharges during July-August over 20,000 m3 s$^{-1}$. More justifications on why choosing July and August 2006 please see our response to the comment" In addition of what you expressed in your answer, your approximation will have to consider a comparison all data from July with those of August between 1999 and 2010.

**Response:**

As suggested, we have included the climatological annual cycle of total river discharges during the years 1999 to 2010 in the Figure 4 of our revised manuscript. The figure together with Figure s1 in the supplement demonstrate that the year 2006 is a wet year and that July and August are among the typical wet season.

**Anonymous Referee #1**

**General Comments**

Congratulations for the research work, there is a large amount of work summarised in a clear and well structured document.

**Specific Comments**

1. Page 4, Line 23: The description of the 1-D model does not refer to any other study, Does this mean that the model was developed for this research? Is there any reference for validating this 1-D model?

**Response:**

The 1-D model was configured and coupled with the 3-D model as detailed in Hu and Li (2009). The 1D-3D coupled model has been validated and applied to study the water-nutrients-sediment budgets (Hu and Li, 2009; Hu et al., 2011), the oxygen budget (Wang et al., 2017), and the nutrient fluxes between the sediment and overlying waters in the Pearl River Estuary (Liu et al., 2016) . In the revised manuscript, we included more details of the 1D-3D coupled model to make it clearer. References for the validation are given at the end of this file.

2. The configuration description of the 1-D model is quite small in contrast with the ones for the 3-D model and the water quality model.

**Response:**

As suggested, we included more descriptions of the 1-D model in the 'Physical model' section in our revised manuscript.

3. Page 7, Lines 13 and 14: mention any quantitative description for the temperature and salinity validation as it is done for the tide.

**Response:**

As suggested, we included some quantitative description for the temperature and salinity validation in our revised manuscript as below:

"The comparisons show small normalized RMSDs (both <0.60 of standard deviations of observations) and high correlations (>0.90 for salinity and >0.80 for temperature) between the model and observations, indicating that the coupled physical model is robust to reproduce the broad-scale features and intra-seasonal patterns of the main hydrodynamic features in the PRE."

More detailed validations can be referred to the Section 3 in our previous study (Wang et al., 2017).

4. Model Validation section: No validation mentioned of the 1-D model

**Response:**

In this study, we use the 1D-3D coupled model with a purpose to account for the interactions of hydrodynamics between the river network and the estuary. The 1D and the 3D model were run in parallel and they exchange model quantities across the coupling interface. The eight outlets (shown in Figure 1 in original manuscript) are the exchange interface of the 1D and 3D models, which serve as the lower boundaries of the 1D model and at the same time the upper boundaries of the 3D model. At each time step, the 3D model utilizes the simulated discharges obtained from the 1D model as the river boundary forcing, and sends the simulated water levels to the 1D model as the downstream boundary forcing as a feedback. Therefore the eight outlets are very important for the assessment of the coupled model performance. We have validated the simulated water levels and/or river discharges against observations at eight outlets in years 1999 (Hu and Li, 2009) and 2006 (Wang et al., 2017 and P7 lines 8-11 in the manuscript). Note that the validations at eight outlets are for both 1D and 3D models. In our revised manuscript, we now provide more details of the 1D-3D coupled model's configuration (in 'Physical model' section) and validations (section 2.1.2) to make it clearer.

5. Section 2.2: Are there any other hypoxia events reproduced by the model throughout the period Nov. 2005 to Dec. 2006, apart of the summer 2006? Are these events observed or not observed? Is the hypoxia event of summer 2006 the only event simulated by the model?

**Response:**

As shown in Figures 4 and 5 in the revised manuscript, our model simulates hypoxia from April to October in 2006. In the simulation, the hypoxia starts to develop in April, peaks in August, and disappears in October.   However, oxygen observations are only available in July and August 2006 when hypoxia are observed, while for other months no observations are available for validating the model simulated hypoxia. This is one of the main reasons why in the manuscript we only focused on July and August in 2006 to study the impacts of riverine inputs on hypoxia and oxygen dynamics in the Pearl River Estuary. Additionally, the previously reported hypoxia also mainly occurred in July and August (Cai et al., 2013). Another motivation of focusing on July and August is that these two months are among the typical wet seasons in the PRE (Figure 4b), which is in line with our study on the effects of riverine inputs. Results of hypoxia for the whole year were added in the section 3.1 in our revised manuscript. Plus, we also provided more justifications on why we focused on July-August only (section 3.1 in the revised manuscript) and added discussions on how representative or comparable was the hypoxia in this time of the year to that of other months or years (section 4.1 in the revised manuscript).

We agree that it would be an interesting topic to study the annual cycle and multi-years variations of hypoxia in the Pearl River Estuary. However, it will be quite hard to study the annual cycle and multi-years variations of hypoxia in this region due to the insufficiency of observational data. And to our knowledge, there are currently few studies on these two topics. Nevertheless, we believe that our study can provide some scientific basis and guidance for further modelling or observational studies on the hypoxia in the Pearl River Estuary.

6. A map showing the location of the study area in a global context will be a great help for the reader which is not familiar with the study area.

**Response:**

We tried as suggested but found it hard to find the Pearl River Estuary in a global map. Alternatively, the location of the Pearl River Estuary in a map of South China Sea is now shown in the revised manuscript.

**Technical Corrections**

• Page 2, Line 17: Why the reference is made on Italic font (Diaz and Rosenberg)

• Not sure which the format for the references within brackets:

Page 2 Lines 19 and 20

Page 5 Line 19

Page 6 Line 8

**Response:**

We have double checked and corrected the format of references throughout the manuscript.

• Page 10, Line 8: where is the definition for **Cont**? I see there are the names for each simulations. Could be possible to mention this before start describing each of them? (The 7

simulations are named as . . . and summarized in table 1)

**Response:**

As suggested, we defined the name for each simulation before describing them in the revised manuscript as below:

"Each group has two simulations, where the concentration of one type of the riverine inputs at eight river outlets is decreased and increased by 50%, respectively. These simulations are named as Base, RivDO-50%, RivDO+50%, RivNtr-50%, RivNtr+50%, RivPOC-50% and

RivPOC+50%, with the basic information of each simulation presented in Table 1."

• Figure 1a [page 24]: It could be just the pdf copy, but the y-axis (latitude) top and bottom labels are missing a 2 (23.5 and 21.5)

**Response:**

We have revised the figure. Please see the Figure 1 in our revised manuscript.

**Suggestions**

• Page 2, Line 13: Enhance instead of exaggerate

**Response**:

We modified this sentence as:

"The classic paradigm for explaining the relations is that excessive nutrient inputs to the coastal oceans stimulate the high primary productivity there, and the subsequent decomposition of the organic matter in the bottom water consumes significant amount of DO that leads to hypoxia."

• Page 5, Line 8: There is a reference for the Mellor-Yamada model but not for the Smagorinsky-type formula. It should be a reference for each of them as they are in the same sentence (line).

**Response**:

Reference has been added in our revised manuscript. The sentence now in the revised manuscript is:

"The horizontal mixing is parameterized by a Smagorinsky-type formula (Smagorinsky, 1963) and the vertical mixing is calculated by the Mellor-Yamada level 2.5 turbulent closure model (Mellor and Yamada, 1982)."

• Page 7 Line 7: Could be possible to specify if the summer is on the north or south hemisphere? It is in the north hemisphere but the suggestion points to be specific as the moths to consider are not the same ones.

**Response**:

We modified the sentence in the revised manuscript as 'The physical-biogeochemical model has been validated against available observations during the July of 1999 in Hu and Li (2009) and July-August 2006 in Wang et al. (2017)'.

• Page 18, Line 4: anthropogenic perturbations instead of just perturbations

**Response**:

Revised as suggested.

**Reference**

Hu, J. and Li, S.: Modeling the mass fluxes and transformations of nutrients in the Pearl River Delta, China, J. Mar. Syst., 78(1), 146–167, doi:10.1016/j.jmarsys.2009.05.001, 2009.

Hu, J., Li, S. and Geng, B.: Modeling the mass flux budgets of water and suspended sediments for the river network and estuary in the Pearl River Delta, China, J. Mar. Syst., 88(2), 252–266, doi:10.1016/j.jmarsys.2011.05.002, 2011.

Liu, D., Hu, J., Li, S. and Huang, J.: Validation and application of a three-dimensional coupled water quality and sediment model of the Pearl River Estuary, Huanjing Kexue Xuebao/Acta Sci. Circumstantiae, 36(11), 4025–4036, doi:10.13671/j.hjkxxb.2016.0145, 2016 (in Chinese with English abstract).

Wang, B., Hu, J., Li, S. and Liu, D.: A numerical analysis of biogeochemical controls with physical modulation on hypoxia during summer in the Pearl River estuary, Biogeosciences, 14(12), 2979–2999, doi:10.5194/bg-14-2979-2017, 2017.

**Anonymous Referee #2**

**Summary:**

In this manuscript the authors use a physical-biogeochemical model to examine a hypoxic event in the Pearl River Estuary (PRE) in July and August 2006. They conduct several numerical experiments in order to determine the relative impact of riverine inputs of oxygen, nutrients and organic matter on hypoxia in the PRE. They specifically examine three processes that affect oxygen dynamics: re-aeration due to air-sea oxygen flux, sediment oxygen demand, and all remaining processes which together is referred to as WCP (water column production). This is a well-written manuscript with some very interesting results, but some clarifications, some more discussion, and a few additional experiments should be performed before publication. The comments below are lengthy, but if addressed fully the resulting paper would be a very valuable contribution to Biogeosciences.

**Major comments:**

1. As I understand it, all results shown in the manuscript are for July and August 2006. This should be made clearer in the abstract, which is written more like this is the "general" case for the PRE. I understand that the model has only been evaluated for July and August 2006, so we don't really know whether the oxygen concentrations at other times of the year are correct or not; however, as a reader I was very interested to see results for the whole summer (May to September), or even for the whole year, rather than just for two months of one year. How does the temporal variability of hypoxia change in the numerical experiments? This analysis does not seem complete without this addition.

Response:

We have made it clearer in the revised manuscript that the results were based on July and August 2006 only. As suggested, results of hypoxia for the whole year were shown in the section 3.1 in our revised manuscript. Plus, we also provided more justifications on why we focused on July-August only (section 3.1 in the revised manuscript) and added discussions on how representative or comparable was the hypoxia in this time of the year to that of other months or years (section 4.1 in the revised manuscript).

In terms of validations, in this manuscript we have only presented the model-data comparison results in July and August 2006. However, the physical-biogeochemical model has been thoroughly validated against not only physical variables (i.e. water levels, salinity, and temperature), DO concentrations, but also the historical observations of some important biological variables (e.g. chlorophyll and particulate organic carbon) and processes (i.e. re-aeration, sediment oxygen demand, and primary productivity) (see Wang et al., 2017). Our model is able to reproduce the observed hypoxia near the Modaomen sub-estuary and the main processes associated with DO. Previous studies have also reported the hypoxia near the Modaomen sub-estuary with the similar spatial extents and characteristics as the model simulated (Cai et al., 2013; Lin et al., 2001; Zhang and Li, 2010).

2. As a reader, I was also wondering whether July and August 2006 was a typical year. Was 2006 a particularly dry July/August? Or wet time period? Are the results of the sensitivity experiments conducted here likely to hold in other years?

Response:

The year 2006 is a wet year with the total river discharge over 10,000 $m^3$ $s^{-1}$ (Figure s1 in the supplement of our revised manuscript) and the monthly river discharges during July-August over 20,000 $m^3$ $s^{-1}$ (Figure 4 in revised manuscript). More justifications on why choosing July and August 2006 are now added to section 3.1 in our revised manuscript. We also added discussions on how representative or comparable was the hypoxia in this time of the year to that of other months or years (section 4.1 in the revised manuscript).

We agree that it would be an interesting topic to study the annual cycle and multi-years variations of hypoxia in the Pearl River Estuary. However, it will be quite difficult to study the annual cycle and multi-years variations of hypoxia due to the insufficiency of observational data. And to our knowledge, there are currently few studies on these two topics. Nevertheless, we believe that our study can provide some scientific basis and guidance for further modelling or observational studies on the hypoxia in the Pearl River Estuary. Guided by this model simulation, we actually have conducted two observation cruises in the Modaomen sub-estuary in January and August this year.

3. One of the main results of this manuscript was that hypoxia in the PRE is not sensitive to nutrient concentrations of the river water entering the region (unlike the Chesapeake Bay and the Gulf of Mexico, for example). This result, however, has to be at least slightly dependent on what value is used for the nutrient concentrations in the eight rivers. What concentrations are used and are they realistic? Where do these concentrations come from? A terrestrial-biogeochemical or watershed model? More detail is needed here. Also, it sounds as if only the nutrient concentrations were changed in the largest river, not all eight rivers. The authors need to show results of changing the concentrations in all eight rivers, not just the largest, since the smaller ones closest to the hypoxic zone might impact the hypoxia zone more than the large river, which is farther from the region of hypoxia. (The same is true for the oxygen and POC experiments.)

**Response**:

As suggested, we have included more details of the riverine inputs at the end of the section 2.1.1 in the revised manuscript:

"River boundary conditions of biogeochemical variables were derived from the monthly observations in 2006 collected by the State Oceanic Administration (including nutrients and DO) and from a previous study (including different classes of dissolved organic carbon, particulate organic carbon, dissolved organic nitrogen, particulate organic nitrogen, dissolved organic phosphorus, and particulate organic phosphorus) (Liu et al., 2016)."

We have also improved the explanations of the numerical experiments setting in the section 2.3 in the revised manuscript.

4. This analysis compares the impact of sediment oxygen demand, re-aeration and WCP on hypoxia. However, this is misleading since WCP is the sum of multiple positive and negative terms. Thus this term is likely smaller than its components. For a more complete analysis, the authors need to separate out the various components of WCP, including respiration, nitrification, water column remineralization etc. . . This is particularly important because in the discussion they state that in the PRE water column respiration/remineralization is not as important as it is in places such as the Chesapeake Bay. But we cannot see this (truly interesting!) result unless the authors isolate these terms.

**Response**:

We understand the reviewer's concerns about the application of water column production (WCP). In our previous study (Wang et al. 2017), we conducted the DO budget analysis and found that the magnitude of nitrification and oxidation are much smaller than the respiration. For the convenience of discussions, we used the water column respiration (WCR, the sum of respiration, nitrification, and remineralization/oxidation) to represent the gross rate of DO consumptions in the water column, a term that has been widely used in the field (Murrell and Lehrter, 2011) and modeling studies (Li et al., 2015; Yu et al., 2015a). According to our budget analysis in Wang et al. (2017), the sediment oxygen demand dominated the DO depletion both for the Pearl River Estuary and the high frequency zone (please see Figure 11a and 12a in Wang et al. 2017), which has been reported by previous studies (Yin et al., 2004;   Zhang and Li, 2010) in other years. In the contrast, the hypoxia in the Chesapeake Bay is dominated by the water column respiration (Li et al., 2015). Differences in the relative importance of water column respiration versus sediment oxygen demand in the two systems (the Pearl River Estuary and the Chesapeake Bay) have been widely accepted by other studies (Rabouille et al., 2008; Hong and Shen, 2013).

In this study, we used the water column production (WCP, the sum of water column respiration and photosynthesis), the re-aeration, and the sediment oxygen demand to represent the net effects of water column, the air-sea interface, and the water-sediment interface, respectively. According to our DO budget analysis in Wang et al. (2017), the photosynthesis and respiration were two major oxygen source or sink term in the water column. Considering that photosynthesis and respiration are both closely and directly correlated to phytoplankton growth, they have the similar distributions and responses to changes in riverine inputs. For example, increasing the nutrient loading will facilitate the growth of phytoplankton and hence both photosynthesis and respiration. Based on these reasons, we did not consider each component separately in our current manuscript.

As suggested, we included more detailed explanations of using the WCP in page 6 of the revised manuscript. Equations for each component of the WCP are now included in the Appendix A of the revised manuscript.

5. It is not completely clear why the "physical modulation" method is needed. If this is a fully coupled physical-biogeochemical model (as is stated), then why can't the authors simply save each of the oxygen flux terms in the oxygen budget? Presenting results in units of DO per unit time (as is done in Figure 7) would be much more helpful for the reader. The idea of different "species" of oxygen seems a bit convoluted. Clearly REA, WCP and SOD have units of oxygen per unit time (see equation 1). Showing figures of these quantities, rather than DO_REA, DO_WCP and DO_SOD would make the manuscript more clearly understandable to readers.

**Response**:

The physical modulation method (now has been renamed as DO species tracing method in our revised manuscript for clarity) was firstly introduced and implemented in our pervious study (Wang et al. 2017) to understand the underlying processes and mechanisms of hypoxia in the Pearl River Estuary. We also conducted DO budget analysis in our previous study and found the advantages of using DO species tracing method in explaining the occurrence, the spatial extent, and the duration of hypoxia in the Pearl River Estuary. Comparing with the budget analysis, the DO species tracing method can demonstrate the spatial connection of each oxygen source or sink process occurring at different locations (e.g. the influence of sediment oxygen demand on adjacent waters and the vertical penetration of re-aeration supplied oxygen).

In the current study, by using the DO species tracing method, we found the buffering effects of re-aeration. That is, the re-aeration can respond to the anthropogenic perturbations of riverine inputs rapidly and hence moderate the DO changes caused by these perturbations. In addition, we further depicted the interactions between each oxygen source and sink processes quantitively. For example, in the Cont simulation (now is renamed as Base simulation), the sediment oxygen demand removed surface oxygen by 2.22 mg $L^{-1}$, which switched the re-aeration from the sink (-1.45 mg $L^{-1}$ day$^{-1}$) to the source (0.55 mg $L^{-1}$ day$^{-1}$) of DO in surface waters.

In our revised manuscript, we have provided more details of the DO species tracing method and made the definition of its associated variables clearer (section 2.2 in our revised manuscript).

6. I really like the idea that re-aerated surface waters can penetrate to the bottom water and offset the changes in DO caused, for example, by increased nutrient, DO, or OM riverine inputs. The authors discuss that this is not the case on the Gulf of Mexico shelf, where hypoxia occurs as a very thin layer near the bottom. The comparison and emphasis on the Gulf of Mexico seems a bit out of place, since the PRE seems to be more similar to the Chesapeake Bay in many ways. The discussion could be strengthened by making a three way comparison between the Chesapeake, Gulf of Mexico and the PRE. Isn't the re-aeration process described here similarly important in the Chesapeake Bay, where hypoxia occurs as a thick layer, which is not far from the surface in a typical July/August?

**Response**:

Firstly, we appreciate reviewer's encouraging comment on our discussion on the re-aeration. However, there seems some misunderstanding in the discussion about the case of the northern Gulf of Mexico (NGOM). In our manuscript, we conducted the comparison between the PRE and the NGOM with a purpose to explain and understand the strong re-aeration in the PRE. As we discussed in our original manuscript (P17 line 11-17), the strong re-aeration is a result of the high sediment oxygen demand and the shallow waters in the PRE. In the contrast, the re-aeration in the NGOM is overall an oxygen sink to surface waters in summer (Yu et al., 2015b) because of the weaker sediment oxygen demand and deeper waters (P17 line 17-23 in original manuscript). However, we did not state that the oxygen supplied by surface re-aeration cannot penetrate to the bottom waters of the NGOM as there is no published studies investigating this mechanism in NGOM. Actually, we think that without applying the species tracing method, it is hard to estimate the effects of surface re-aeration on the bottom waters because it depends highly on the magnitude of re-aeration, the water depth, and the hydrodynamic conditions.

Secondly, we agree that it will be very interesting to discuss the role of re-aeration on hypoxia in other hypoxic systems (e.g. the NGOM and the Chesapeake Bay) and make the comparisons with the PRE. However, extended discussion is not feasible due to a lack of data and relevant studies in other hypoxic systems. Take the Chesapeake Bay as an example, we don't have re-aeration data to figure out the importance of re-aeration in the Chesapeake Bay. As suggested, we included some explanations for why we did not include the Chesapeake Bay into discussions and our speculations of re-aeration in the Chesapeake Bay in the section 4.3 of our revised manuscript:

"In the other hypoxic system, the Chesapeake Bay as described earlier, extended discussion on the importance of re-aeration is limited by a lack of observations and relevant studies of re-aeration. Nevertheless, according to our results, we can speculate that the re-aeration might be quite important in the Chesapeake Bay because the strong water column respiration can draw down the surface DO concentrations and enhance the re-aeration. However, the penetration of the oxygen supplied by re-aeration to the bottom layer is hard to be estimated without applying the DO species tracing method like our study or method similar in the Chesapeake Bay. In general, more relevant studies are required to examine the role of the re-aeration on hypoxia in the Chesapeake Bay."

7. There is some considerable overlap with the authors' previous publication (Wang et al.,2017, BG). For instance, it appears to me that one of the main points in the abstract of the current manuscript: "Model results showed that hypoxia in the Pearl River Estuary was confined to the shelf off the Modaomen sub-estuary with a hypoxic area of 200km2 mainly due to the combined effect of re-aeration and sediment oxygen demand" was actually a primary result of this previous publication. This should be made clearer in the abstract and introduction. Clearly this study builds off the previous study. Although the previous study is mentioned in the abstract, the differences between the current study and the previous study should be made clearer to the readers.

**Response**:

As suggested, we have clarified the differences between the current and previous studies in the revised manuscript.

**Minor Comments:**

8. Abstract last sentence – suggest changing this to: "This study highlights the importance of re-aeration in determining the hypoxic extent and in reducing hypoxia variability in shallow estuaries."

**Response**:

Revised as suggested. We changed the sentence as below:

"This study highlights the importance of re-aeration in reducing hypoxia variability in shallow estuaries."

We removed "the hypoxic extent" because it is the conclusion from our previous study (Wang et al. 2017).

9. Abstract – Define here (and in introduction) that by re-aeration you mean a flux of oxygen across the air-sea interface. (Currently this doesn't occur until page 6).
**Response**:
    Revised as suggested. The re-aeration is defined in the abstract and introduction in the revised manuscript as below:

(In *Abstract*) "Changes in the riverine inputs of DO and nutrients had little impacts on the simulated hypoxia because of the buffering effects of re-aeration (DO fluxes across the air-sea interface)"

(In *Introduction*) "A more recent study by Wang et al. (2017) further points out that the balance of oxygen in the PRE is mainly controlled by the source and sink processes occurring in local and adjacent waters, among which the re-aeration (DO fluxes across the air-sea interface) and SOD determine the spatial distributions and durations of hypoxia in the PRE."

10. Introduction – Authors could mention climate change as another anthropogenic impact, since recent studies are showing that increasing temperatures have a large impact on increasing hypoxia.
**Response**:
    Revised as suggested. We mentioned climate change in Introduction of the revised manuscript as follow:

"Recent years have also seen an increasing number of studies showing that climate variation contributes to the spreading hypoxia in coastal oceans. The climate variation can change the ocean circulation or the vertical stratification to alter the balance between the oxygen source and sink processes (Rabalais et al., 2010). A modeling study conducted in the Chesapeake Bay has shown the good correlations between the climate variation, stratification, and the observed DO (Du and Shen, 2015). In addition, the global warming, as a symptom of climate variation, is another factor that can enhance the hypoxia. For example, Laurent et al. (2018) predicted a prolonged and more severe hypoxia in the northern Gulf of Mexico under a projected future (2100) climate state where the global warming leads to reduction in oxygen solubility and increased stratification."

11. P2, line 7: Why is there a ten-year lag? Does this occur in an estuary like the PRE? Or maybe it's not relevant here.
**Response**:
The ten-year lag between eutrophication and hypoxia is estimated on the global scale (Rabalais et al., 2010). There hasn't been study on the time lags between the eutrophication and hypoxia in the Pearl River Estuary. We have deleted this sentence in our revised manuscript.

12. Page 4, line 11: This paragraph is talking about how nutrient inputs to the Pearl River Estuary can impact hypoxia, but this line is about particulate organic carbon, which could be moved to the following paragraph talking about organic matter. Page 4, line 16: What is the organic matter? Is it only POC mentioned in line 11? Or does it include PON (nitrogen) and dissolved organic matter? Which type of organic matter primarily contributes to hypoxia?
**Response**:
As suggested, we moved this sentence to the next paragraph describing particulate organic carbon in our revised manuscript. In P4 line 16 of the original manuscript, organic matter represents the particulate organic carbon only. We have clarified it in our revised manuscript. Now the sentence in the revised manuscript is:

"In addition to the nutrient loading, the particulate organic carbon (POC) are another important form of anthropogenic inputs that influence the hypoxia in the estuary (~2.5×10$^6$ t yr$^{-1}$ from the Pearl River network (Zhang et al., 2013))."

13. P4, line 20: How are these models dynamically coupled? If these were dynamically coupled, the estuarine model would provide feedbacks to the riverine model. Is that the case? Also, the model set up seems to assume that there are no freshwater or nutrient sources (from the land) into Mirs Bay, Daya Bay or Honghai Bay. Is there evidence to support this assumption?

**Response:**

The 1D river network model and the 3D estuary model are dynamically coupled through the eight river outlets. These two models are run in parallel and their model quantities are exchanged across the coupling interface (eight outlets) during runtime. At each time step, the 3D model utilizes the simulated discharge obtained from the 1D model as the river boundary forcing, while the 3D model sends the simulated water levels to the 1D model as the downstream boundary forcing for the next time step. More detailed descriptions of the coupling can be seen in Hu and Li (2009) and we have included more details in the "Physical model" section of our revised manuscript.

The riverine input of freshwater and nutrient fluxes entering the Mirs Bay, Daya Bay, and Honghai Bay are much lower than those from the Pearl River Network. In addition, the Mirs Bay, Daya Bay, and Honghai Bay are quite far away from the hypoxic zone. In the wet season, the Pearl River Estuary is dominated by the southwesterly monsoon and the Pearl River plume mainly propagate eastward. Therefore, we neglected freshwater or nutrient sources entering the Mirs Bay, Daya Bay, and Honghai Bay, and also neglected contributions of these regions to the oxygen dynamics in hypoxic zone.

14. P5, "Water quality model" section: In this section the authors need to describe more clearly where their riverine biogeochemical concentrations are derived from, since these are at the very heart of their numerical experiments. Do concentrations of the 26 state variables all come from the riverine model described above? If so, more information regarding the details of the biogeochemistry of the riverine model is needed. Where do the outer boundary conditions come from, for the estuarine model? How about atmospheric deposition of nutrients, like nitrate and phosphate? Are all these assumed to be negligible? How realistic is this assumption?

**Response**:

We added the following in the "water quality model" section (renamed as "biogeochemical model section" in our revised manuscript):

"River boundary conditions of biogeochemical variables were derived from the monthly observations in 2006 collected by the State Oceanic Administration (including nutrients and DO) and from a previous study (including different classes of dissolved organic carbon, particulate organic carbon, dissolved organic nitrogen, particulate organic nitrogen, dissolved organic phosphorus, and particulate organic phosphorus) (Liu et al., 2016). The open boundary conditions of biogeochemical variables were specified following Zhang and Li (2010)."

In this manuscript, we did not include the atmospheric deposition of nutrients because the riverine nutrient input is the dominant nutrient source in the Pearl River Estuary.

15. P6, line 1: Since one of the conclusions of the manuscript is the relative importance of SOD compared to WCP (see abstract), here the terms making up "WCP" need to be written out explicitly.

**Response**:

There might be some confusion over whether the conclusions are from our previous study (Wang et al. 2017) or this study, which we have better clarified in the revised manuscript. The relative importance of SOD (sediment oxygen demand) and WCP (water column production) is one of the conclusions from our previous study (Wang et al. 2017), but not in current study. This study builds on Wang et al. (2017) to focus on the response of each oxygen source and sink process to the different riverine inputs and their impacts on the hypoxia. We did emphasize the importance of SOD and the re-aeration to hypoxia in this manuscript but only aim to remind readers some key features of hypoxia formation in the Pearl River Estuary before more extensive investigations through sensitivity experiments.

Following the suggestion, equations of each component of WCP and some relevant descriptions have been added in the Appendix A of our revised manuscript.

16. P6, line 7: Please provide the equations for photosynthesis, respiration, nitrification and oxidation (potentially in an appendix), and provide values of all parameters used. (The reference used here for the model is a white paper from 14 years ago. The model has been adjusted since then. Are the authors really using those original parameters and equations? Please include information on the version of the model that is being implemented.)

**Response**:

      As mentioned in our response to comment #4, we provided the equations for each component of WCP and the descriptions of relevant parameters in the Appendix A of our revised manuscript. The reference here (HydroQual, 2004) is the manual of the RCA. Equations we used in this manuscript are the same as in this document. Parameters are set according to this document and previous studies (Liu et al., 2016; Zhang and Li, 2010) that used the same physical-biogeochemical model for the Pearl River Estuary. Values of the primary parameters have been summarized in Table A2 in our revised manuscript.

17. P6, line 12: Define what is meant by "dissolved matter". Is this dissolved organic matter, i.e. DON and DOC? Or dissolved nutrients, i.e. ammonium? Or both?

**Response**:

      We think here the reviewer meant P6, line 27. Here the 'dissolved matter' is referred to all nutrients and DO considered in our model. We have clarified it in our revised manuscript as below:

"In the RCA, a sediment flux module is incorporated to simulate the depositional flux of particulate organic matter (i.e. particulate organic carbon, particulate organic nitrogen, and particulate organic phosphate), the diagenesis processes in the sediment, and the transport of nutrients and DO from the sediment to the overlying water (Figure 2). Detailed descriptions about the sediment flux module can be seen in the Appendix B."

18. P6, line 18: As above, please provide values of these parameters within this paper (possibly in an appendix.)

**Response**:

      Values of primary parameters have been provided in Table A2 of our revised manuscript.

19. P6, line 27: As above, please provide equations and parameter values for DO_sed (possibly in an appendix)

**Response**:

As suggested, we have provided equations of DO_sed and descriptions about our sediment flux module in the Appendix B of our revised manuscript.

20. P7, line 2: Earlier the authors stated that this is a dynamically coupled model, but here it sounds as if the water quality model is being run offline from the physical model, which would indicate that there is no dynamic coupling, and the biological simulation cannot impact the physics. Please make it clearer in the text as to whether the models are truly dynamically coupled, or simply run offline.

**Response**:

True, the physical model and the water quality model are only one-way coupling, where the physical model can affect the water quality model but the water quality model cannot impact the physical model. As suggested, we have revised the model description in our manuscript.

20. P 7 line 4: What data is being referred to here and how was it used? Data assimilation? Forcing? Validation?

**Response**:

Here we refer to the data used as the model input (e.g. riverine input and open boundary conditions) and for the model validation. We removed this sentence in the revised manuscript and instead we gave more detailed descriptions as below:

"Initial conditions were obtained from a two-month spin-up simulation which was repeated for three times to reach a steady state. River boundary conditions of biogeochemical variables were derived from the monthly observations in 2006 collected by the State Oceanic Administration (including nutrients and DO) and from a previous study (including different classes of dissolved organic carbon, particulate organic carbon, dissolved organic nitrogen, particulate organic nitrogen, dissolved organic phosphorus, and particulate organic phosphorus) (Liu et al., 2016). The open boundary conditions of biogeochemical variables were specified following Zhang and Li (2010)."

21. P7, line 7: There are actually very few observations of DO presented in Wang et al. (2017). Are these really the only observations available of DO in the PRE region? Is nothing more available since 2006? It also looks like the oxygen data shown in Wang et al. rarely, if ever, actually go hypoxic?

**Response**:

    During 2006, we only have observations in July and August. However we do have collected and analyzed the observation data from 1993 to 2009 (Figure r3), finding that there are very limited observations near the Modaomen sub-estuary (where the high frequency zone is located). Nevertheless, the model simulated hypoxia near the Modaomen sub-estuary in this study has also been reported in previous observational (Cai et al., 2013; Lin et al., 2001) and modelling studies (Zhang and Li, 2010), and the simulated spatial extent and characteristics here are consistent with those from previous studies. Additionally, noticing that the available oxygen data for validating simulated hypoxia is insufficient, in Wang et al. (2017) we have thoroughly validated the model against not only physical variables (i.e. water levels, salinity, and temperature), DO concentrations, but also the historical observations of some important biological variables (e.g. chlorophyll and particulate organic carbon) and processes (i.e. re-aeration, sediment oxygen demand, and primary productivity).

    For the observational data in Wang et al. (2017), the minimum observed DO concentrations are below the hypoxic level (defined as 3 mg $L^{-1}$ here) and the observed hypoxic area is about 150 $km^2$ (please see Figure r1). It should be noted that the hypoxic area shown in Figure r1 was estimated based on the observation available at limited time and space and might not fully represent the true state of hypoxia in the entire Pearl River Estuary.

[Figure]

Figure r1. Multi-years variations in the observed minimum DO concentration (left panel) and
hypoxic area. Red cycles indicate the July-August 2006.

22. P8, line 3: Because the authors have only evaluated model results for oxygen in July and
August 2006, does this mean these results only are valid for that year? Is that a particularly
wet year or a dry year? Or an average year? Can you put this year in perspective? (Perhaps in
the discussion?)
**Response**:
Please see our response to comment #2.

23. Page 8, line 13: This sentence seems to indicate that this estimation is not straightforward
only in river dominant estuaries. How about tide dominant estuaries, which can also be
impacted by local and remote source and sink processes? P9, line 8: What does "Cont" stand
for? Continuous? I would think "Base" or "Reference" or "Realistic" might be better
descriptions of this simulation.
**Response**:
The estimation is not straightforward in the tide dominant estuaries either. According
to our results, the oxygen supplied by re-aeration can penetrate to the bottom waters through
the vertical diffusion (see section 4.1 in Wang et al. (2017) and also P15 line 13-18 in the original manuscript). It follows that in a tide dominant estuary, the tide-induced mixing can facilitate the penetration. In the revised manuscript, we have modified the sentence as below:

"However, in a river and tide dominated estuary such as the PRE, this estimation is not straightforward because of the spatial connections of each source and sink process occurring in different locations."

The 'Cont' meant a control case. As suggested, we have renamed 'Cont' as 'Base' simulation in our revised manuscript.

24. Section 2.3: The text is not clear here. Are the concentrations of DO and nutrients reduced in all 8 rivers, or only the Humen? Also it is not clear whether the concentrations of DO and nutrients in the experiments are set to what is predicted in 2050, or are simply increased by 50%. In reality, the concentrations in 2050 will depend on management decisions which are very difficult to predict. I think it's best to state here that you increased/decreased the concentrations by 50%, and if you want to convince the reader that these are representative of 2050 and 1970 respectively, then bring this up in the discussion. Please provide the concentrations of DO and nitrate (as an example nutrient) used in each of these experiments. More detail is needed here. If freshwater flows stay the same, this should be stated.

**Response**:

The riverine inputs of DO, nutrients, and particulate organic carbon are reduced in all 8 rivers. In RivNtr+50% simulation, we increased the nutrient loading by 50% in all 8 rivers, which is close to the increase in nutrient loading in 2050 predicted by Strokal et al., (2015). As suggested, we have provided more details about the numerical experiments in the section 2.3 of our revised manuscript.

25. P11, line 19: Remove HFZ acronym since it is not used elsewhere. Please define the hypoxic frequency zone more quantitatively since this is used throughout the text. Where exactly is this? It's hard for the reader to know. Does it change in time?

**Response**:

We removed the HFZ acronym in the revised manuscript as suggested. The high frequency zone is defined in page 13-14 in our revised manuscript as below:

"The high frequency zone here is defined as the area encompassed by the 10% isoline of July-August averaged hypoxic frequency and is denoted by the white contour in Figure 8."

We have depicted the high frequency zone with white lines in Figure 8 in the revised manuscript. The high frequency zone remains unchanged in time in our model simulation.

26. P11, line 23: The word "additionally" should come before "occurs" since hypoxia also occurs on the shelf.

**Response**:

Revised as suggested.

27. P13, line 9: Also list percent changes in hypoxia area and volume, as was done above.

*Response:*

Revised as suggested. In page 14 of the revised manuscript, we say:

"As a result, the hypoxic area and hypoxia volume only increase by about 10% in the RivNtr-50% simulation in relative to the Base simulation"

28. P13, line 22: Considering using PRE acronym earlier. (It hasn't been used much since very early in the manuscript.)

**Response**:

Revised as suggested.

29. P14, line 14: Aren't there two POC simulations/experiments, not three?

**Response**:

Here we meant the two POC simulations and the Cont simulation (renamed as Base simulation as suggested). As shown in table 1 in our revised manuscript, we have two POC simulations with increased or decreased riverine inputs of particulate organic carbon by 50%. We have clarified it in our revised manuscript as below:

"The two POC simulations and the Base simulation have identical physical processes and hence same temperature limitation."

30. P14, line 24: In the results, it would make sense to discuss Figure 7 (the "Cont" results) before the sensitivity experiment results, rather than inside the section 2.3 sensitivity experiment section.

**Response**:

We think the reviewer meant section 3.3 instead of section 2.3 of the original manuscript here. In the section 3.3 of our original manuscript, we found that changing the riverine inputs of particulate organic carbon could affect the re-aeration by changing the sediment oxygen demand and water column production (WCP). This finding leads to the discussion of Figure 7 (Figure 9 in the revised manuscript) which is to demonstrate the mechanisms of how the sediment oxygen demand influence the surface re-aeration quantitatively. Therefore we would like to keep the current presentation flow.

31. P14, line 25: The figure shows 0.53, not 0.55?

**Response**:

Right the value should be 0.53. We have corrected it in the revised manuscript as below:

"First, the SOD consumes bottom DO by 0.53 mg $L^{-1}$ $day^{-1}$ and decrease the upward advective DO fluxes reaching the upper layer by 0.34 mg $L^{-1}$ $day^{-1}$."

32. P15, line 1: How does the reader compute 0.13 from Figure 7?

**Response**:

We have more detailly described the calculation in our revised manuscript as below:

"As shown in Figure 8, the SOD can affect the DO concentrations in the upper layer indirectly through the interactions with the vertical advection, the vertical diffusion, and the horizontal advection as explained below. First, the SOD consumes bottom DO by 0.53 mg $L^{-1}$ $day^{-1}$ and decrease the upward advective DO fluxes reaching the upper layer by 0.34 mg $L^{-1}$

day$^{-1}$. Second, the deoxygenation induced by SOD can increase the vertical DO gradient and facilitate the downward vertical diffusion of oxygen by 0.02 mg L$^{-1}$ day$^{-1}$ from the upper layer. Finally, the decreased upper DO concentrations affect the horizontal outfluxes of DO and ultimately result in a higher net horizontal advective flux by 0.21 mg L$^{-1}$ day$^{-1}$. Consequently, the net effect of the SOD on the upper DO is 0.15 mg L$^{-1}$ day$^{-1}$. which causes a decline of 2.22 mg L$^{-1}$ in DO concentrations in the surface layer. ''

In our original manuscript, we ignored the interactions between the sediment oxygen demand and the vertical diffusion to get 0.13 mg L$^{-1}$ day$^{-1}$.

33. P15, line 2: Based on equation 8, I would think the dark blue DO bar would equal the sum of all the other bars, but this doesn't seem to be the case? Why is this?

**Response**:

The sediment oxygen demand (SOD) is a sink for the oxygen and therefore there is a negative sign in front of the DO$_{SOD}$ in equations (1), (3), (7)-(9), and (13) in the section 2.1 and 2.2. However, in the other sections we did not include the negative sign in DO$_{SOD}$, where DO$_{SOD}$ has positive value and represents the amount of oxygen removed by the sediment oxygen demand. Accordingly, the equation should be $\Delta DO = \Delta DO_{BC} + \Delta DO_{REA} + \Delta DO_{WCP} - \Delta DO_{SOD}$ (Dark blue bars = light blue bars + green bars + orange bars - yellow bars in the Figure 5 b-c). We have corrected the equations (1), (3), (8)-(9) in the revised manuscript to be consistent for the sign of DO$_{SOD}$.

For example, compared with the 'Cont' simulation (now renamed as the Base simulation), decreasing the riverine inputs of particulate organic carbon in RivPOC-50% will weaken the SOD and lead to a decrease in the magnitude of DO$_{SOD}$ (the removal of oxygen caused by the SOD). That means the bottom DO will be increased by 0.51 mg L$^{-1}$ (decrease in DO$_{SOD}$ but increase in DO). In addition, decreasing the POC inputs weakens the light attenuation and facilitates the primary productivity, leading to an increase of bottom DO by 0.3 mg L$^{-1}$ (increase in DO$_{WCP}$). At the same time, the decrease in riverine POC inputs will weaken the re-aeration and decrease the bottom DO by 0.22 mg L$^{-1}$ (increase in DO$_{REA}$). Combining all of these processes, the bottom DO will be elevated ultimately by 0.51+0.3-0.22=0.59 mg L$^{-1}$.

34. P15, line 11: It is important to qualify the 217km2 statistic by saying that this is true only for a July/August average in 2006. This is not true for other months of the year, and we don't know whether this is true for other years.

**Response**:

    We removed this sentence in the revised manuscript. We have clarified that this result is based on the July-August 2006 elsewhere in our revised manuscript.

35. P15, line 15: DO_REA is not a term that your readers will be familiar with (unless they have read this paper carefully). This paper will have a greater impact if this could be reworded such that processes are mentioned, i.e. discuss the re-aeration of surface water via air-sea flux (in units of oxygen per unit time), rather than DO_REA.

**Response**:

    The text in P15, line 15 of the original manuscript meant the oxygen supplied by the re-aeration can penetrate to the bottom waters and compensate the oxygen loss caused by other processes.   In our original manuscript, we did not use the word 're-aeration' (in units of oxygen per unit time) because re-aeration is the air-sea flux of oxygen occurring at the air-sea interface, which does not guarantee to penetrate to the bottom waters.

    We agree that the term $DO_{REA}$ is not familiar to readers, so we defined the key variables more clearly (e.g. $DO_{BC}$, $DO_{WCP}$, $DO_{SOD,}$ and $DO_{REA}$) in the revised manuscript and list their definition in Table A1 in the revised manuscript.

36. P15, line 19: Again where is the hypoxic frequency zone? Where is "the west of the lower estuary"? Also, make it clearer that this is a result of Wang et al. (2017) and not of this paper.

**Response**:

    Please see the response to comment 25 for the high hypoxic zone. Here we have clarified that this is the result based on Wang et al. (2017) and we moved this sentence to section 4.3 in our revised manuscript. The west of the lower estuary is represented by the red box in the new Figure 8d.

37. P15, lines 4-8: This is a very interesting result! But unfortunately this paper does not show any statistics on water column respiration, so this is not clear. Please separate out the various terms inside WCP so the reader can see specifically that water column respiration is
not large here.
**Response**:
 Please see our response to the comment 4.
38. P16, line 15: I don't think the authors mean the residence time of the Mississippi River,
which extends a great distance, well up into the continent of North America. Do you mean
the shelf plume area? This section would be much stronger if the authors compared all three
systems mentioned here: the GoM, Chesapeake Bay and the PRE.
**Response**:
Here we meant the residence time of bottom waters in the hypoxic zone of the shelf.
The residence time of 95 days is cited from the Rabouille et al., (2008) where they compared
the hypoxia in four different river systems (i.e. The Yangtz river, the Mississippi River, the
Pearl River, and the Rhone River).
We have compared the three hypoxic systems (i.e. the Chesapeake Bay, the northern
Gulf of Mexico (NGOM), and the PRE) to explain why the hypoxia in PRE is most sensitive
to the riverine input of particulate organic carbon (P16 line 3-18 of original manuscript). The
main differences between the three hypoxic systems are summarized in Table r1, which has
been included as the Table 3 in the revised manuscript. As we discussed in the manuscript, in
contrast to the NGOM and the Pearl River Estuary, the water column respiration induced by
excess nutrients is the dominant oxygen depletion process in the Chesapeake Bay. As a result,
the hypoxia in the Chesapeake Bay is very sensitive to the nutrient loading. In the NGOM
and the PRE, the dominant roles of the sediment oxygen demand have been reported in many
previous studies based on both observational (Murrell and Lehrter, 2011; Yin et al., 2004)
and modelling studies ((Yu et al., 2015b; Zhang and Li, 2010). Hypoxia in the NGOM can be
well simulated with appropriate parameterization of SOD while neglecting the water column
processes (Yu et al., 2015a). However, in the NGOM, the particulate organic carbon (POC)
produced by phytoplankton (autochthonous POC) is the major contribution to the sediment
oxygen demand. While in the PRE, the riverine input of POC (allochthonous POC) is the
dominant source. The differences in relative contributions of allochthonous POC versus autochthonous POC between the NGOM and the PRE are thereafter discussed in the revised manuscript.

Table r1. A summary of characteristics of hypoxia among three systems (i.e. Chesapeake Bay, northern Gulf of Mexico, and Pearl River Estuary)

| | WCR dominant | SOD dominant | |
| | | autochthonous POC dominant | allochthonous POC dominant |
| --- | --- | --- | --- |
| **Chesapeake Bay** | ✓ | | |
| **Northern Gulf of Mexico** | | ✓ | |
| **Pearl River Estuary** | | | ✓ |

39. P16: Rather than discussing terrestrial vs. marine POC, I think it would be clearer to discuss autocthonous vs. allochthonous POC. "Marine POC" sounds as if it comes from outside the hypoxic zone from the ocean, but I don't think this is what is meant?

**Response**:

We have revised the terms as suggested in our revised manuscript. We had attempted to use the terrestrial POC and marine POC to represent the POC delivered from the river network and produced by the phytoplankton, respectively.

40. P16, line 22: Is July-August a wet or dry season?

**Response**:

The July-August are typical wet seasons with the monthly averaged river discharges over 20,000 m$^3$ s$^{-1}$. We have stated it in the section 3.1 of the revised manuscript to explain why we focused on July-August 2006:

"Another motivation of focusing on July and August is that these two months are among the typical wet seasons in the PRE (Figure 4b), which is in line with our study on the effects of riverine inputs."

41. Section 4.1: This section needs to describe more completely the difference in marine vs. terrestrial POC in the PRE vs. Gulf of Mexico. Why does terrestrial POC not impact hypoxia? Just because that the POC entering from the river is relatively small and sinks out before making it al the way to the shelf? Or is there something specifically different about the terrestrial matter entering from the Mississippi compare to that being delivered to the PRE? Is this a residence time issue? Is the terrestrial source more important in the PRE because the nutrient inputs are quite low, compared to what they are in the Gulf of Mexico? What about in the Chesapeake Bay?

**Response**:

More discussions on the differences in the PRE and the Northern Gulf of Mexico (NGOM) have been included in the section 4.2 of the revised manuscript:

"The different POC sources in the NGOM and the PRE might be explained by their distinct physical and biogeochemical processes (Table 4). Firstly, the relative magnitudes of autochthonous versus allochthonous POC are different in the two hypoxic systems. The allochthonous inputs of POC in the NGOM and PRE are at the same magnitude: $3.8 \times 10^6$ t $yr^{-1}$ (Wang et al., 2004) and $2.5 \times 10^6$ t $yr^{-1}$ (Zhang et al., 2013), respectively. However, the autochthonous inputs in the two systems are different. According to our model results, the primary productivity in the PRE is $310.8 \pm 427.5$ mg C $m^{-2}$ $day^{-1}$, which is within the range of $183.9 \sim 1213$ mg C $m^{-2}$ $day^{-1}$ reported by Ye et al., (2014). However, the observed primary productivity in the NGOM ranges from 330 to 7010 mg C $m^{-2}$ $day^{-1}$ (Quigg et al., 2011), the upper range of which is much higher than that in the PRE. The relatively lower primary productivity in the PRE is a result of the stronger phosphorus limitation (DIN:DIP ratio of 126 in the PRE versus 33 in the NGOM, respectively) and the light shading effects of high suspended sediment concentrations. The dominant role of the allochthonous POC in highly turbid estuaries have been reported in previous studies (Fontugne and Jouanneau, 1987; Middelburg and Herman, 2007). Secondly, fates of the allochthonous POC in the two systems are different due to the difference in the residence time between the systems. In the

PRE, the residence time is 3~5 days during the wet season, which is much shorter than in the NGOM (~95 days). It follows that the allochthonous POC cannot be degraded completely and hence can significantly fuel the SOD in the PRE. The difference in surface salinity distribution can also be used to explain the different relative roles of allochthonous POC in the two hypoxic systems. Previous studies have suggested a good correlation between the relative contributions of allochthonous POC and the salinity, namely the contributions of allochthonous POC generally decrease as salinity increases seaward (Fontugne and Jouanneau, 1987; Middelburg and Herman, 2007). Similar correlations have also been reported in the PRE (Yu et al., 2010) and NGOM (Wang et al., 2004). The surface salinity in the high hypoxia frequency zone varies between 0 to 10 psu during the wet season based on our model results, while the surface salinity in the hypoxic zone of the NGOM is saltier than 24 psu even in the wet season according to the results from a well-validated physical model in Yu et al. (2015a). This implies a more important role of allochthonous POC in the PRE than in the NGOM. Finally, compositions of the allochthonous POC are different in the two hypoxic systems. Zhang and Li (2010) mentioned that contributions of labile POC to the allochthonous POC are higher in the PRE than in the NGOM."

Unlike the PRE and the NGOM, the hypoxia in the Chesapeake Bay is more controlled by the water column respiration instead (Hong and Shen, 2013). And the hypoxia is sensitive to the nutrient loading. It follows that the autocthonous POC is more important than the allochthonou POC to the hypoxia formation. One possible reason is probably the relatively long residence time in the Chesapeake Bay (180 days (Du and Shen, 2016)), which allows the complete degradation of the allochthonou POC before entering the hypoxic zone."

42. P17, line 17: Isn't the same likely to occur in Chesapeake Bay? This might be a very interesting discussion point here.
**Response**:

Here we compared the Pearl River Estuary with the northern Gulf of Mexico to demonstrate that the high re-aeration in the Pearl River Estuary is due to the high sediment oxygen demand and shallow waters. Extended discussions on the Chesapeake Bay are largely limited by the lack of observations and relevant studies on the re-aeration. To our knowledge, few studies have mentioned the direction (source vs. sink) or the magnitude of re-aeration in the Chesapeake Bay. According to our results, we can speculate qualitatively that the strong water column respiration may enhance the re-aeration in the Chesapeake Bay.

As suggested, we have included a brief discussion on the Chesapeake Bay at the end of the section 4.3 of our revised manuscript.

**Figures:**

43. Figure 1: The figures do not look to be italicized (as it says in the caption). Fig 1b does not add any significant information to what appears in Fig 1a. In Fig 1c "grids" should be "grid". The Fig 1a caption should note that this is a bathymetric map.

**Response:**

We have modified the caption in the revised manuscript.

We would like to keep Figure 1b that shows the computational cross-sections of 1-D model. In the revised manuscript, we have stated the purposes of showing Figure 1a and Figure 1b.

We have replaced 'grids' with 'grid' in Figure 1c and revised the caption of Figure 1a as suggested.

44. Figure 3: The text refers to 3a and 3b, but the left panel is not marked (b), and there is no reference to (b) in the caption.

**Response:**

We have added the missing (b) in the left panel. We also added the reference to (b) in Figure 3.

Figure 5: The y-axes in (b)-(d) should specify that these are "Changes in concentration", not concentrations themselves. Also, (b)-(d) should have same y-range to make it easier for the reader to compare all three figures.

**Response:**

Revised as suggested.

45. Figure 6: Please label figures (a)-(e) and provide captions for each. What is the white line? Axes are not labeled.

**Response**:

  We have provided captions for each panel in the revised manuscript. The white line in Figure 6 (new Figure 8 in revised manuscript) denotes the high frequency zone. We also added the information in the revised manuscript. Labels have been added to the axes in Figure 6.

46. Figure 7: This figure is a little confusing, because one would expect that the vertical diffusion out of box 1 would represent the vertical diffusion into box 2. I gather the net diffusion arrows are shown, but maybe it would make more sense to show the middle layer as having a +0.02 diffusion into the middle layer at the top, and a -0.17 diffusion out of the middle layer at the bottom? But why doesn't this equal 0.48? Maybe I'm confused because this is only DO_SOD, and not total oxygen? Wouldn't this be a more enlightening figure if all the DO fluxes were shown here?

**Response**:

  Yes, fluxes shown in Figure 7 (new Figure 9 in the revised manuscript) represent the net fluxes of -$DO_{SOD}$. The purpose of Figure 7 is to explain how the sediment oxygen demand can affect the surface re-aeration. Other DO fluxes can be seen in Figure 11 and Figure 12 in our previous study (Wang et al., 2017). In our revised manuscript, we made a statement of the purpose of Figure 7 and referred other DO fluxes to Wang et al. (2017).

  As suggested, we also modified the Figure 7 to show the influx and outflux separately. The new figure which will be used in the revised manuscript is shown here. The negative values represent the outfluxes and the positive values represent the influxes. Take the vertical diffusion (red arrows) as an example, the outflux from the upper layer is 0.02 mg L$^{-1}$ day$^{-1}$. Considering the mass conservation, the mass of DO (in unit of mg) leaves from the upper layer should be equal to the mass enters the lower layer. As we mentioned in our original manuscript (P14 line 22-23), volume of the middle layer is three times as large as the upper and bottom layers. Therefore, the influx into the middle layer should be 0.02/3≈0.01 mg L$^{-1}$ day$^{-1}$. As analogy to the upper-middle layer, the outflux from the middle layer should be one thirds of the influx into the bottom layer (0.48/3≈0.16 mg L$^{-1}$ day$^{-1}$). Since the influx and outflux of the middle layer are -0.01 and 0.16 mg L$^{-1}$ day$^{-1}$, respectively, the net flux is 0.15 mg L$^{-1}$ day$^{-1}$ as was shown in the Figure 7 in our original manuscript.

[Figure]

Figure r2. Budget of $-DO_{SOD}$ for the upper layer, middle layer, and bottom layer in the Pearl River Estuary for the 'Cont' case (has been renamed as Base case in the revised manuscript).

**English language comments:**

Throughout, "organic matters" should be changed to "organic matter". And similarly "dissolved matters" should be "dissolved matter".

P 4, line12 – processes should be process

P6, Line 26 – transportation should be transport

P7, line 7 – delete "here" and "as"

P8, line 16 – should be "interacting"

Page 12, line 1: "further" should be "farther"

P14, line 17: should be " layer, exerting a strong constraint"

P15, line 14: supply should be supplies

P15, line 24: most should be "more"

P16, line 5: should be "are the most important processes"

**Response**:

Thank you for the detailed comments. We have modified them as suggested in our revised manuscript.

[revised manuscript text omitted]